# A single-cell atlas of lung homeostasis reveals dynamic changes during development and aging
Hao Jia [1,6], Yuan Chang [1,6], Yulin Chen[1,2,6], Xiao Chen[1], Hang Zhang[1], Xiumeng Hua[1], Mengda Xu[1], Yixuan Sheng[1], Ningning Zhang[1], Hao Cui[1], Lei Han[1,3], Jian Zhang [4,7] ✉, Xiaodong Fu [5,7] ✉ & Jiangping Song [1,7] ✉

Aging is a global challenge, marked in the lungs by function decline and structural disorders, which affects the health of the elderly population. To explore anti-aging strategies, we develop a dynamic atlas covering 45 cell types in human lungs, spanning from embryonic development to aging. We aim to apply the discoveries of lung's development to address aging-related issues. We observe that both epithelial and immune cells undergo a process of acquisition and loss of essential function as they transition from development to aging. During aging, we identify cellular phenotypic alternations that result in reduced pulmonary compliance and compromised immune homeostasis. Furthermore, we find a distinctive expression pattern of the ferritin light chain (*FTL*) gene, which increases during development but decreases in various types of lung cells during the aging process.

Aging poses a global challenge as human longevity increases and the population's age structure changes[1]. Despite longer life spans, the proportion of disease-free life time has not kept pace, with 16%-20% of seniors suffering from late-life illness and organ failure[2,3]. The lung, with its large functional surface and complex cellular composition, is subjected to various noxious stimuli in the aging process[4]. The aging lung experiences reduced gas exchange and immune capacity along with structural changes including airway remodeling and decreased pulmonary compliance. These changes in the aging lung are the driving factors of lung failure and susceptibility to respiratory diseases[5–7].

Aging is the result of multiple factors, and treatment targets have been explored for various mechanisms of aging[8]. However, these anti-aging therapies are facing clinical challenges, including target non-specificity, clinical safety risks, and the difficulty of translating animal models to human treatment. The anti-aging approaches based on the theory of heterochronic parabiosis have achieved a breakthrough, using the circulatory medium of young individuals to resist the aging of organs and tissues[9,10]. Based on this theory, we believe that the distinct gene expression patterns of the embryo, the most vigorous life cycle, can provide insights to solve the aging problem.

Given the complexity of lung cell components, single-cell RNA sequencing (scRNA-seq) is an ideal method for resolving gene expression patterns in aging and embryonic lungs.

In this study, we built a consecutive atlas of human lung development and aging. Results from scRNA-seq showed dynamic changes in component cells and gene expression patterns. We identified genes, such as ferritin light chain (*FTL*), that undergo specific expression change patterns, offering a potential biomarker and therapeutic target for aging lungs.

## Results

### Total cell populations in the developing and aging lungs

The lungs of aborted fetuses and adult lungs were collected and divided into five groups (gestational week or age dependent)[11,12], including the first-trimester group (T1, *n* = 2), the second-trimester group (T2, *n* = 3), the last-trimester group (T3, *n* = 1), the non-aging adult group (T4, *n* = 2), and the aging adult group (T5, *n* = 2). Sample information can be found in Supplementary Table 1. The total pulmonary tissue cells in the five groups were analyzed (Fig. 1a). Quality control filtered out nFeature_RNA < 200 and percent.mt > 10 cells (Supplementary Fig. 1a, b), and captured the

[1]Beijing Key Laboratory of Preclinical Research and Evaluation for Cardiovascular Implant Materials, Animal Experimental Centre, National Centre for Cardiovascular Disease, Department of Cardiac Surgery, Fuwai Hospital, Chinese Academy of Medical Sciences and Peking Union Medical College, Beijing, China. [2]Department of Traditional Chinese and Western Medicine, Gansu University of Chinese Medicine, Lanzhou, China. [3]Department of General Surgery, Yanan Hospital, Kunming Medical University, Kunming, China. [4]Thoracic Surgery Department, the third affiliated hospital of Sun Yat-sen University, Sun Yat-sen University, Guangzhou, China. [5]Department of Cardiology, Guangzhou Institute of Cardiovascular Disease, Guangdong Key Laboratory of Vascular Diseases, the Second Affiliated Hospital of Guangzhou Medical University, Guangzhou, China. [6]These authors contributed equally: Hao Jia, Yuan Chang, Yulin Chen.[7]These authors jointly supervised this work: Jian Zhang, Xiaodong Fu, Jiangping Song. ✉e-mail: sumszhangjian@163.com; fuxiaod@gzhmu.edu.cn; fwsongjiangping@126.com

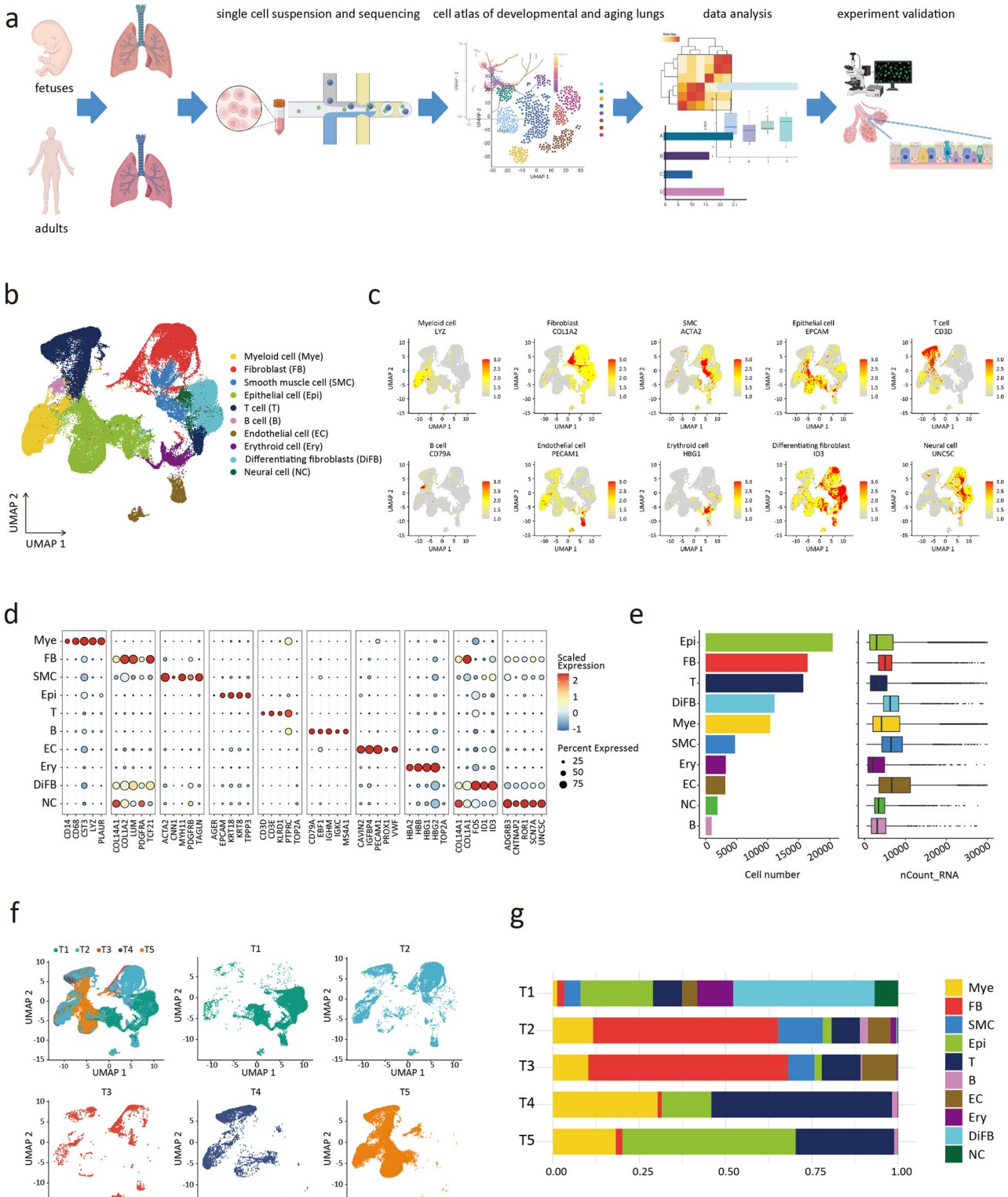

**Fig. 1 | A single-cell atlas of the developing and aging human lungs. a** Study flowchart. **b** UMAP plots showing the main cell cluster distribution of the developing and aging human lungs. **c** Expression of classical marker genes used to define the main cell clusters. **d** Dot plots showing the specific marker genes of the main cell clusters. **e** The number of cells of each main cell cluster. **f** UMAP plots showing the cell distribution in 5 phases. **g** For each of the 5 phases, the fraction of cells originating from each of the 10 main cell clusters.

transcriptional profiles of 88,055 cells (T1 group: 26863; T2 group: 24488; T3 group: 3707; T4 group: 6839; T5 group: 26158). The canonical correlation analysis (CCA) method was used to adjust the batch effect. An unbiased clustering and uniform manifold approximation and projection (UMAP)

analysis was used to identify 29 cell clusters with distinct cellular transcriptomic signatures (Supplementary Fig. 2a). The 29 cell clusters were assigned to 10 main cell clusters, which were annotated as myeloid cell (Mye), fibroblast (FB), smooth muscle cell (SMC), epithelial cell (Epi), T cell

(T), B cell (B), endothelial cell (EC), differentiating fibroblast (DiFB), neural cell (NC), and erythroid cell (Ery) (Fig. 1b). Different main cell clusters were characterized with specific marker genes and gene expression patterns[13], such as *LYZ* (Mye), *COL1A2* (FB), *ACTA2* (SMC), *EPCAM* (Epi), *CD3D* (T), *CD79A* (B), and *PECAM1* (EC) (Fig. 1c). Since the 2 clusters of fibroblasts had distinct differentiating characteristics, DiFB was described separately (Fig. 1d and Supplementary Fig. 2b).

In the fetal lungs, the FB/DiFB cluster represented the most substantial population at stages T1, T2, and T3. In contrast, epithelial and immune cells were the dominant cell populations in adult lungs (Fig. 1e, f, g, and Supplementary Fig. 2c, d). Our discovery in the cell proportion of human fetal lungs was similar to that of fetal rodents[14,15]. The main cell cluster profile of adults observed in our study was consistent with prior reports[16].

## Dynamic changes of pulmonary epithelial cells during development and aging

In aging lungs, pulmonary dysfunction is characterized by altered gas exchange and decreased mucociliary clearance[17], suggesting a critical role of epithelial cells in the aging process. We detected 20458 epithelial cells and clustered them into 8 subclusters. Among these cell subclusters, alveoli Epi include alveolar epithelial type 1 cell (AT1) and alveolar epithelial type 2 cell (AT2), and airway Epi include ciliated cell (Cil), club cell, pulmonary neuroendocrine cell (PNEC), basal cell (Bas), goblet cell (Gob), and reactive cell (Fig. 2a and Supplementary Data 1). We found a continuous decrease in the Bas ratio in the proportion of airway Epi (Fig. 2a and Supplementary Data 2). The relative loss of Bas as progenitor cells during aging might be involved in airway regeneration and repair disorders[18]. The proportion of Cil showed a unimodal trend (Fig. 2a). Each subcluster had a distinct gene expression profile (Fig. 2b)[19,20].

In terms of transcriptional noise[21,22], we found that both AT1 and AT2 experienced a decrease in transcriptional noise during development, indicating an enhancement in transcriptional stability. During aging, the transcriptional noise of AT1 and AT2 increased, and the efficiency of transcriptional stability, accuracy, and mature mRNA decreased (Fig. 2c). This trend was not observed in the airway Epi.

To perform unsupervised clustering of genes with similar expression change patterns during development and aging, we used the Time Course Sequencing Data Analysis (TCA) method. In the AT2 subcluster, the TCA results of cluster 4 reached the peak of the gene expression score in the T3 group (Supplementary Fig. 3a), with the up-regulated gene patterns in development. Gene Oncology (GO) analysis suggested that the genes up-regulated during development were functionally enriched in ribosome-related protein synthesis (Fig. 2d). Similarly, cluster 1 of AT1 showed the genes up-regulated during development were enriched in protein synthesis (Fig. 2g and Supplementary Fig. 3b). To analyze the expression changes of AT2 aging genes, we combined the TCA method (cluster 5) and the analysis of T5/T4 DEGs. GO analysis revealed that AT2 had reduced antigen processing and presentation ability, decreased surfactant homeostasis, and increased gene expression associated with immune cell chemotaxis during aging (Fig. 2e, f). For AT1, the gene expression related to cell adhesion and morphogenesis was down-regulated (Fig. 2h), while the gene expression of immune cell chemotaxis was up-regulated during aging (Fig. 2i). The transcriptional profiles of AT1 and AT2 during development and aging were similar, which suggested that AT2 retained age-related transcriptional features when it differentiated into AT1[23].

Among the DEGs in alveoli Epi, *FTL* was the most significantly differentially expressed gene between groups. *FTL* was up-regulated during development and down-regulated during aging (Fig. 2j, l), which was confirmed by immunostaining results (Fig. 2k, m, and Supplementary Fig. 4a, b). The function of *FTL* was to maintain the homeostasis of intracellular iron[24,25]. Lung iron homeostasis is closely related to oxygen sensing, pathogen defense, and chronic respiratory diseases[26]. Combined with our findings, we suggested that FTL was a marker and potential intervention target for lung aging. The result of RT-qPCR supported our discovery (g. 3f). In addition, we explored another distinctively expressed gene, Eukaryotic

Translation Elongation Factor 1 Alpha 1 (*EEF1A1*), in AT1. *EEF1A1* was up-regulated in both late embryos and aging adults (Fig. 2n, o, and Supplementary Fig. 4c). It is associated with the translation of viral proteins and viral replication in severe acute respiratory syndrome coronavirus 2 (SARS-CoV-2)[27]. We hypothesized that the relative susceptibility of the elderly and infants to COVID-19 may be related to the high expression of EEF1A1[28,29], which suggested that the antiviral drug plitidepsin may have an ideal effect on elderly patients and infants[30]. However, this speculation was based on the drug target. Due to the unique pharmacokinetics in infants, we could not give definitive clinical suggestions based on this study. We reconstructed alveoli Epi relationships by pseudotemporal trajectory (Fig. 3a). The results of trajectory feature and GO terms were consistent with the findings of TCA and DEGs analysis. We also found the regular expression changes of *FTL* in the trajectory (Fig. 3b).

We analyzed the dynamic changes of airway Epi transcription profiles. For Cil, cluster 1 gene sets were enriched in cilium movement and microtubule-based movement (Fig. 3c and Supplementary Fig. 3c), which suggested that the structure and function of Cil progressively matured during development. During aging, the expression of genes related to the inflammatory process was up-regulated in Cil, while the expression of genes related to cell adhesion and epithelial cell migration was down-regulated (Fig. 3d, e). Considering the progenitor cell identity of Bas, we conducted TCA and DEGs analysis of Bas. GO terms suggested that the homeostasis and metabolism patterns matured during development, which were lost in aging and were accompanied by decreased cell adhesion and upregulation of apoptosis levels (Fig. 3f,–g,–h, and Supplementary Fig. 3d). These results revealed that in addition to the changes in cell number distribution during development and aging, there were dynamic changes in cell function-related transcription profiles of Cil and Bas[31]. The function of Club cells is to protect airway epithelium, detoxify, and regenerate into Cil[32]. TCA and DEGs analysis suggested that the proliferation of club cells was decreased and immune-related processes were up-regulated during aging (Fig. 3i,–j,–k, and Supplementary Fig. 3e). The pseudotime trajectory analysis showed the transition from development to aging over the time course (Supplementary Fig. 5a, b).

Collectively, we summarized the transcriptional pattern in Epi and found that the cell function of each subcluster, such as protein synthesis for AT2 and cilium movement for Cil, gradually matured during development. This maturation process began in the fetus, preparing for normal respiratory function after birth. The above cell functions decreased during aging, with the upregulation of inflammatory and immune-related processes. In addition, we found that stem cell depletion in airway Epi, including Bas and club cells, was a continuous process from development to aging, which could lead to susceptibility to age-related chronic lung diseases[33].

## Phenotypic changes of fibroblasts during development and aging

FB is involved in the morphogenesis of the lung during development[34], and during aging, lung compliance decreases, in which FB is involved in the process of interstitial remodeling[35]. To investigate dynamic changes in FB phenotype and gene expression pattern, we captured 27511 FBs and DiFBs and divided them into 9 subclusters, including the alveolar FB, differentiating FB, universal FB, myofibroblast, adventitial FB, peripheral nerve FB, and pericyte (Fig. 4a). Each cluster had a specific gene expression pattern (Fig. 4b and Supplementary Fig. 6a)[36,37]. Alveolar FB was divided into two clusters, and alveolar Fibs 1.2 (*SCN7A⁺ GRIA1⁺*) showed that they had the potential to be excited by glutamatergic signaling inputs[16]. We analyzed the transcriptional noise of FB and found that the transcriptional noise of FB increased during aging (Fig. 4c).

We explored the dynamic changes of FB transcriptional profiles. Gene clusters 2 and 6 of FB were enriched in epithelial tube morphogenesis, cell junction assembly, mesenchyme development, and extracellular matrix (ECM) organization (Fig. 4d, e, and Supplementary Fig. 6b), which matched the role of FB in development[34]. During aging, GO terms of DEGs between T4 and T5 groups suggested that FB in the aging group was characterized by

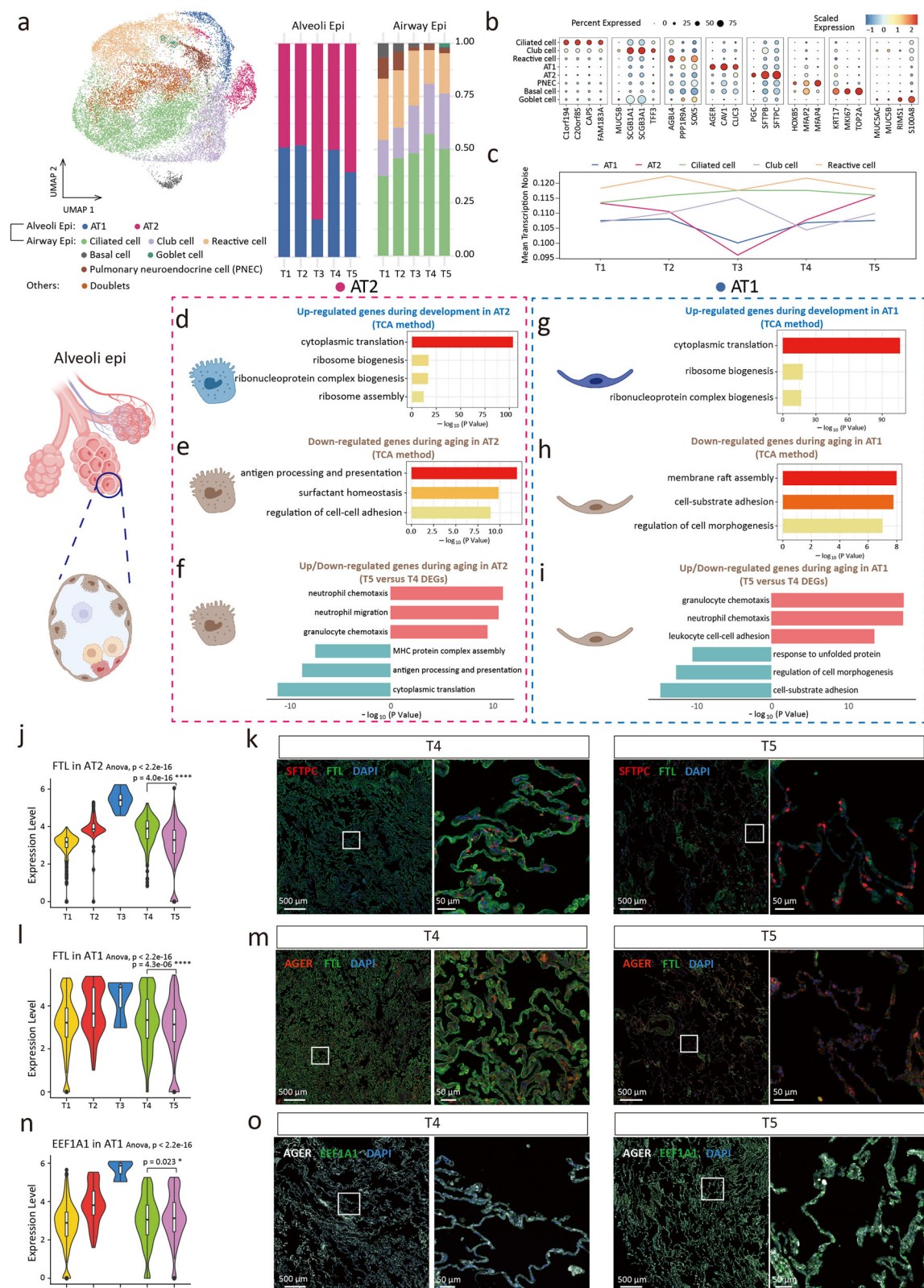

increased immune cell chemotaxis and disordered ECM organization (Fig. 4f). Our evaluation of elastin synthesis and collagen catabolism in FB showed that elastin synthesis and collagen catabolism were reduced in the aging group (Fig. 4g, h)[38]. According to previous reports[39], these phenotypes were associated with decreased lung compliance. Masson staining suggested an increased fibrosis level in the aging lung (Fig. 4i). We collected cohort data on lung function and found that vital capacity (VC), forced vital capacity (FVC), and forced expiratory volume in one second (FEV1)/FVC were significantly reduced in the aging adult group (Table 1), supporting our findings of FB phenotypic changes[40]. A similar change pattern of *FTL* was found in FB, aligned with the immunofluorescence results (Fig. 4j, k, and Supplementary Fig. 6c).

**Fig. 2 | Dynamic changes of epithelial cell during development and aging.**
**a** UMAP plots of epithelial cells (left). Alveolar epithelial type 1 cell (AT1); Alveolar epithelial type 2 cell (AT2); Pulmonary neuroendocrine cell (PNEC). Proportion of epithelial cell subclusters in 5 groups (right). **b** Dot plots of specific marker genes. **c** The age-associated changes of transcriptional noise of epithelial cell subclusters. **d** Gene ontology enrichment analysis of gene set in cluster 4 of AT2 (development) by TCA method. **e** Gene ontology enrichment analysis of cluster 5 of AT2 (aging). **f** Gene ontology enrichment analysis of up (top part, red)/down (bottom part, blue)-regulated genes expression in aging AT2. **g** Gene ontology enrichment analysis of gene expression in cluster 1 of AT1 (development). **h** Gene ontology enrichment

analysis of gene expression in cluster 3 of AT1 (aging). **i** Gene ontology enrichment analysis of up (top part, red)/down (bottom part, blue)-regulated genes expression in aging AT1. **j** Relatively expression of *FTL* in AT2 subcluster in single-cell analysis. **k** The expression of FTL in AT2 of immunofluorecence staining. **l** Relatively expression of *FTL* in AT1. **m** The expression of FTL in AT1 of immunofluorecence staining. **n** Relatively expression of *EEF1A1* in AT1. **o** The expression of EEF1A1 in AT1 of immunofluorecence staining. The choice of ANOVA tests was based on the results of Shapiro-Wilk normality test. Dunnett tests were used to inter-group comparison. $*p < 0.05$, $****p < 0.0001$.

Our SCENIC analysis on FB showed that Zinc Finger Protein 217 (*ZNF217*) was up-regulated in FB of the aging group (Supplementary Fig. 6d, e). The upstream positive regulator of *ZNF217*, Metastasis Associated Lung Adenocarcinoma Transcript 1 (*MALAT1*)[41], was highly expressed in the senescent FB (Supplementary Fig. 6f). Thus, the *MALAT1-ZNF217* regulatory pathway might be involved in the cellular senescence and phenotypic changes of FB in aging lungs[42].

Collectively, the organization of ECM and the morphogenesis function of FB increased throughout development. During aging, phenotypic changes of FB contributed to reduced pulmonary compliance.

### Dynamic changes of smooth muscle cells and endothelial cells
As the main components of the vascular system, EC and SMC contribute to the development and aging of the lungs. During development, EC directs the differentiation of lung stem cell[43]. During aging, EC and SMC mediate the pathological progression of pulmonary hypertension[44,45].

We captured 4716 SMCs in 3 clusters, including vascular smooth muscle cell (VSMC), airway smooth muscle cell (ASMC), and dividing SMC (Fig. 5a, b, and Supplementary Fig. 7a). VSMCs ($NTRK3^+MEF2C^+$) and ASMCs ($MYLK^+HHIP^+$) had distinct gene expression profiles[46,47]. The phenotypic transition regulates the structural and physiological profile of VSMCs[48,49]. Using the TCA method, gene clusters 4 and 5 were enriched in muscle contraction, muscle cell development, and aerobic metabolism during development (Fig. 5c, d, and Supplementary Fig. 7b).

A total of 3155 ECs were captured and clustered, including capillary EC, arterial EC, venous EC, lymphoid EC, and dividing EC (Fig. 5e, f, and Supplementary Fig. 8a). Capillary ECs were divided into general capillary EC ($BTNL9^+IL7R^+$) and airway capillary EC ($HPGD^+TBX2^+$)[50]. Changes in gene expression of EC indicated that regulated genes in capillary EC, arterial EC, and venous EC were enriched in endothelium development (Fig. 5g, h, and Supplementary Fig. 8b,–c,–d,–e,–f,–g). Analysis of gene expression of capillary EC during aging found the down-regulated genes were related to the nitric-oxide synthase biosynthetic process, while the up-regulated genes were related to SMC proliferation regulation and cell adhesion (Fig. 5i). Given the endothelial phenotypic changes as typical characteristics of pulmonary hypertension[51,52], we performed phenotypic scores across groups (Fig. 5j and Supplementary Fig. 8h). The level of EC proliferation level decreased during development, but no differences were observed between T4 and T5, and EC apoptosis showed a similar trend (Fig. 5j). This suggested that the changes in EC during aging were mainly at the transcriptional profile rather than relative cell number. Changes in the transcriptional characteristics of EC during aging might be involved in pulmonary vascular remodeling[53], which was related to the susceptibility of elderly people to pulmonary oxygen exchange disorder[54].

Collectively, the contractile function of SMC has been enhanced over the course of development. For EC, loss of transcriptional identity and phenotypic change were involved in the remodeling of aging pulmonary vessels.

### Changes in immune function and cell proportion of lymphoid cells
A total of 16687 lymphoid cells (Lym) were collected and clustered into 12 subclusters. The clusters included NK cell, proliferating lymphocyte, tissue-resident memory CD4$^+$ T cell (CD4$^+$ T$_{RM}$), effector memory CD8$^+$ T cell (CD8$^+$ T$_{EM}$), naïve T cell, immune response T cell, B cell, differentiating T cell, dividing T cell, CD4$^+$ Treg cell, mast cell, and lymphoid lineage derived dendritic cell (Fig. 6a, b, and Supplementary Fig. 9a)[55–57]. Changes in the cell

distribution ratio indicated that the proportion of NK cell and naïve T cell increased during development. During aging, the percentage of NK cell and naïve T cell decreased (Fig. 6a). The decreased proportion of naïve T cells was one of the characteristics of immunosenescence[58,59]. We observed an increased tendency for transcriptional noise during aging in the Lym subclusters (Fig. 6c).

In the analysis of gene expression changes in Lym, we focused on naïve T cell and NK cell. Our finding indicated during development, genes up-regulated in naïve T cells were enriched in biological processes of T cell activation, differentiation, and proliferation (Fig. 6d and Supplementary Fig. 9b), which were related to T cell function and proliferation. Genes down-regulated in aging were enriched in cytoplasmic translation and T cell activation (Fig. 6e). The results suggested that the renewal capacity and immune function of naïve T cells decreased during aging. Analysis of T4 and T5 DEGs indicated that genes related to the regulation of immune cell chemotaxis and the adhesion of leukocyte cells were strongly expressed during aging. Genes related to cytoplasmic translation and T cell differentiation were down-regulated (Fig. 6f), which was consistent with the results of the TCA analysis. In NK cells, genes that were up-regulated during development were associated with leukocyte-mediated cytotoxicity, T cell activation, and cell killing (Fig. 6g and Supplementary Fig. 9c), indicating an increased NK cells' immune activity. During aging, both TCA and DEGs analysis indicated genes highly expressed in NK cells were related to immune cell chemotaxis, whereas genes related to T cell activation, antigen processing and presentation, and NK cell-mediated immunity were down-regulated (Fig. 6h, i). The results of the pseudotime trajectory analysis of T cells showed that the differentiation level of T cells increased during development (Supplementary Fig. 9d). This confirmed that the differentiation and maturation of T cells began before birth[60].

Collectively, the proportion of subclusters and transcription profile of Lym showed a dynamic change during development and aging.

### Changes in the immune function of myeloid cell and cell-cell interaction
For Mye, we captured 10383 cells and clustered these cells into 7 subclusters, including tissue-resident macrophage, M1-like macrophage, M2-like macrophage, monocyte-derived macrophage, dendritic cell, neutrophil, dividing Mye (Fig. 7a, b, and Supplementary Fig. 10a)[61–63]. The relative proportion of Mye subclusters suggested that the proportion of tissue-resident macrophage increased in adult lungs, while the proportion of dividing Mye decreased as development progressed (Fig. 7a). The above trend of cell proportion change was consistent with the findings of Li et al.[64]. Attention was paid to the transcriptional noise of Mye, and we found that transcription noise increased with aging, except for M1-like and M2-like macrophages (Fig. 7c).

Through TCA analysis, we found that up-regulated genes during development were enriched in cytokine production and myeloid cell homeostasis (Fig. 7d and Supplementary Fig. 10b). Analysis of TCA and DEG for aging suggested that genes regulating chemotaxis and migration were highly expressed, and the expression of genes involved in detoxification and antigen processing and presentation was down-regulated (Fig. 7e, f). Due to the unique role of tissue-resident macrophage in development and aging[65,66], we examined the gene expression alternations within this subcluster. The TCA analysis showed that, during development, the up-

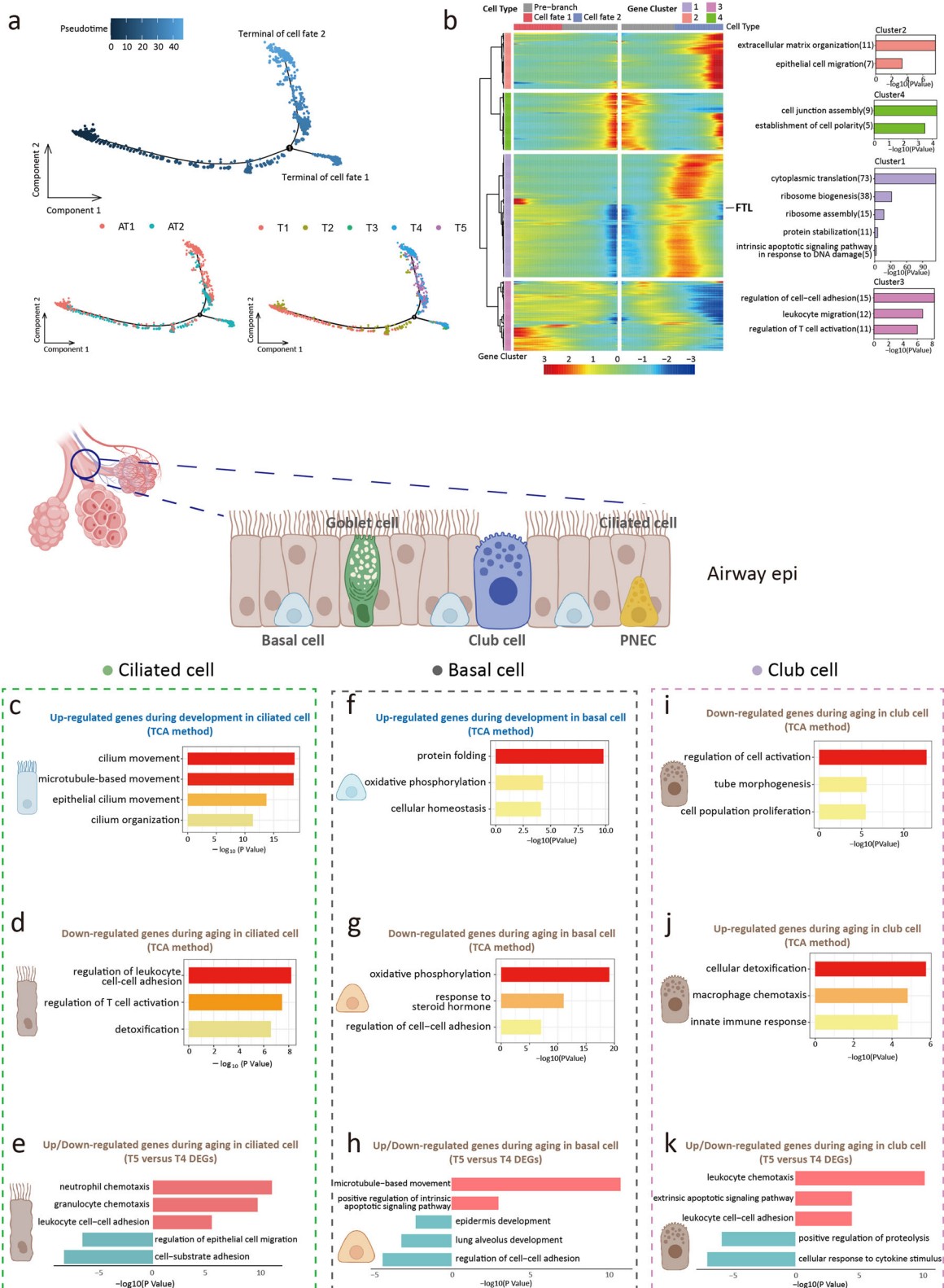

**Fig. 3 | Pseudotime trajectory analysis and dynamic changes of epithelial cell during development and aging. a** Pseudotime trajectory of alveoli epithelial cells inferred using monocle2 method. **b** Heatmap showing differentially expressed genes (DEGs) alone with the pseudotime as in (**a**), catalogs into 4 gene module clusters (left). Gene ontology terms enriched for each gene module clusters (right). **c** Gene ontology enrichment analysis of gene expression in cluster 1 of ciliated cell (development). **d** Gene ontology enrichment analysis of gene expression in cluster 6 of ciliated cell (aging). **e** Gene ontology enrichment analysis of up (top part, red)/down (bottom part, blue)-regulated gene expression in aging ciliated cell. **f** Gene ontology enrichment analysis of gene expression in cluster 2 of basal cell (development). **g** Gene ontology enrichment analysis of gene expression in cluster 6 of basal cell (aging). **h** Gene ontology terms of up (top part, red)/down (bottom part, blue)-regulated gene expression in aging basal cell. **i** Gene ontology enrichment analysis of gene expression in cluster 1 of club cell (aging). **j** Gene ontology enrichment analysis of gene expression in cluster 6 of club cell (aging). **k** Gene ontology terms of up (top part, red)/down (bottom part, blue)-regulated gene expression in aging club cell.

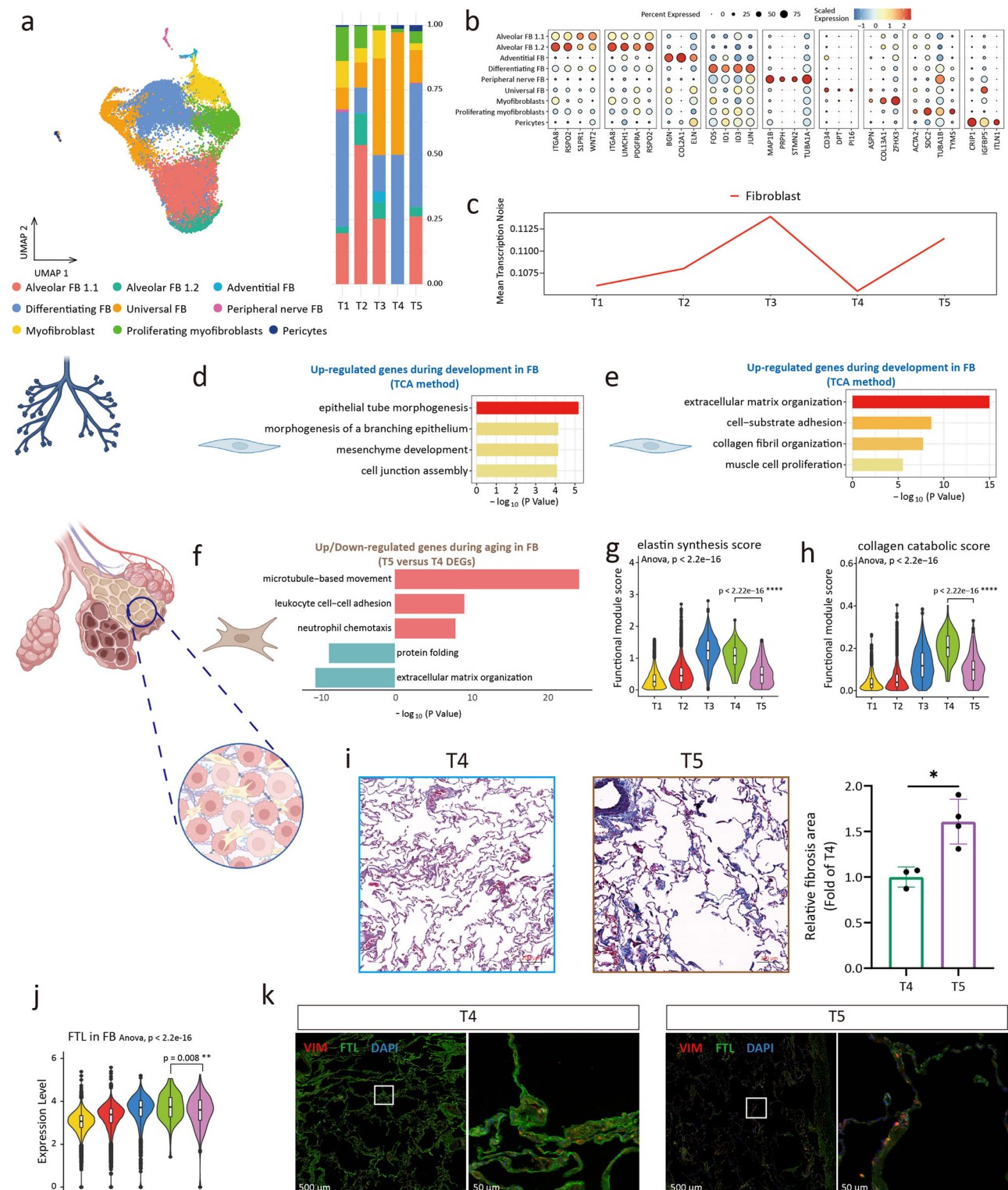

**Fig. 4 | Dynamic changes of fibroblast during development and aging. a** UMAP plots of fibroblast (FB) (left). Proportion of fibroblast subclusters in 5 groups (right). **b** Dot plots of specific marker genes of fibroblast. **c** The age-associated changes of transcriptional noise of fibroblast. **d** Gene ontology enrichment analysis of gene expression in cluster 2 of fibroblast (development). **e** Gene ontology enrichment analysis of gene expression in cluster 6 of fibroblast (development). **f** Gene ontology enrichment analysis of up (top part, red)/down (bottom part, blue)-regulated gene expression in fibroblast. **g** Elastin synthesis (GO: 0071953) score of fibroblasts between 5 groups. **h** Collagen catabolic (GO: 0030574) score of fibroblasts between 5 groups. **i** Masson-stained sections of lung tissues between 5 groups and relative fibrosis area of the lung tissues shown as the means ± SD, tested by two-tailed $t$-test. *$p < 0.05$. **j** Relatively expression of *FTL* in fibroblast in single-cell analysis. **k** The expression of FTL in fibroblast of immunofluorecence staining. The choice of ANOVA tests was based on the results of Shapiro–Wilk normality test. Dunnett tests were used to inter-group comparison. **$p < 0.01$, ****$p < 0.0001$.

**Table 1 | Lung function characteristics in non-aging and aging adults**

| Variables | Non-aging adults (n = 18) | Aging adults (n = 16) | P-value |
|---|---|---|---|
| Age (years) | 45.00 (43.00–49.00) | 68.50 (66.00–74.25) | <0.0001# |
| Male (n, %) | 3 (16.67) | 8 (50.00) | 0.088§ |
| FVC (forced vital capacity) (L) | 3.17 ± 0.58 | 2.50 ± 0.64 | 0.003* |
| FEV1 (forced expiratory volume in one second) (L) | 2.55 (2.34-2.80) | 1.78 (1.50-2.51) | 0.002# |
| FEV1/FVC (%) | 81.72 ± 5.43 | 77.50 ± 6.37 | 0.045* |
| VC (vital capacity) (L) | 3.07 ± 0.56 | 2.47 ± 0.64 | 0.007* |
| FEV1/VCmax (%) | 81.50 (77.50-84.25) | 77.00 (72.25-82.50) | 0.156# |
| PEF (peak expiratory flow) (L/s) | 5.99 ± 1.57 | 4.69 ± 1.51 | 0.020* |
| FEF50% (forced expiratory flow 50%) (L/s) | 3.65 (2.90–4.48) | 2.22 (1.71–3.20) | 0.004# |
| PIF (peak inspiratory flow) (L/s) | 3.07 (2.38–4.09) | 2.07 (1.26–2.65) | 0.004# |
| TV (tidal volume) (L) | 1.14 ± 0.40 | 1.08 ± 0.36 | 0.669* |
| IRV (inspiratory reserve volume) (L) | 1.11 ± 0.58 | 0.85 ± 0.46 | 0.159* |
| ERV (expiratory reserve volume) (L) | 0.87 ± 0.31 | 0.63 ± 0.38 | 0.051* |
| RR (respiratory rate) (1/min) | 18.54 ± 4.72 | 18.35 ± 4.58 | 0.904* |
| MVV (maximal voluntary ventilation) (L/min) | 108.30 ± 22.58 | 68.57 ± 20.64 | <0.0001* |

Depending on the normality, lung function characteristics of non-aging and aging adults were presented by mean ± SD or median (IQR). Normality was tested by Shapiro–Wilk test. Homogeneity of variance was tested by Variance Ratio test.

P-value: * Two-sided t-test; # Mann–Whitney test; § Chi-square incorporating Yates' correction for continuity. A two-sided p-value of <0.05 was considered statistically significant.

regulated genes of the tissue-resident macrophage were enriched with phagocytosis, cell killing, and defense response. These results suggested that the immune function of tissue-resident macrophage gradually matured during development (Fig. 7g and Supplementary Fig. 10c). During aging, chemotaxis and migration genes were highly expressed, and genes related to detoxification, T-cell activation, myeloid activation, and antigen processing and presentation were down-regulated (Fig. 7h, i). We found that hotspot genes associated with age-related pulmonary diseases (pulmonary hypertension, pulmonary emphysema, COPD, and asthma) were highly enriched in Mye and Epi (Supplementary Fig. 10d).

The cellPhoneDB results showed that the interaction between the main cell clusters increased during development and declined during aging (Fig. 8a). However, the interaction between Mye and other cell populations remained elevated during aging (Fig. 8a), which suggested that the immune-related processes were relatively active during aging.

We then focused on the cell-cell interactions of the AT2 and AT1 subcluster. We found that in AT2 and AT1, the interaction scores of AT2-Mye and AT1-Mye through SCGB3A1-MARCO were significantly increased, especially in tissue-resident macrophage (Fig. 8b, c). This observation aligned with prior findings that identified the ligand-receptor relationship SCGB3A1 (UGRP1)-MARCO as a facilitator of pulmonary inflammation. In addition, this prior study suggested that this ligand-receptor interaction occurred primarily through tissue-resident alveolar macrophage, which was similar to the findings of our study[67].

In terms of cytokine activation and signaling, we observed a decrease in the expression level of CISH within both Epi and Mye clusters. CISH functioned as a suppressor of the cytokine signaling system (Supplementary Fig. 11a)[68], and its reduced expression disrupted the negative feedback regulatory circuits, which was associated with chronic pulmonary inflammation[69]. Correlation analysis suggested that CISH was significantly correlated with cellular senescence (Supplementary Fig. 11b).

**Decreased FTL expression induced cellular senescence**

The cell homeostasis of the aged lung decreased, and our single-cell atlas of development and aging lungs suggested that the dysregulation of iron homeostasis caused by FTL down-regulation might be involved in this process. The decrease in FTL affects the ferritin ensemble. Correlation analysis suggested that FTL expression level in AT2 subcluster was positively correlated with the intracellular iron storage pathway (P-value < 0.0001;

cor = 0.865) (Fig. 9a). The decrease of FTL expression led to an increase in the level of intracellular free iron ions, and these excess iron ions disrupted redox homeostasis[70–72]. Correlation analysis also found that FTL expression level was negatively correlated with cellular senescence (P-value < 0.0001; cor = −0.436) (Fig. 9b). We used siRNA transfection system to verify the relationship between FTL expression level and cellular senescence. Aimed at exploring the roles of FTL in cellular senescence of lung Epis, we used siRNA transfection system to treat BEAS-2B cells. Through RT-qPCR and Western Blot analysis, siRNA-FTL#3 showed reliable knockdown ability (Fig. 9c and Supplementary Fig. 12). Considering the relationship between cell vitality and cellular senescence[73], we used Cell Counting Kit-8 (CCK-8) to detect the effect of FTL on cell vitality. Compared to the control group, the cell vitality of FTL-knockdown group was significantly decreased (Fig. 9d). We next performed the cellular senescence assay, SA-β-gal staining results showed that BEAS-2B cells were induced to senescence after siRNA treatment (Fig. 9e). Moreover, BEAS-2B cells treated with siRNA-FTL showed increased expression of p21 (Fig. 9f and Supplementary Fig. 13).

The results of scRNA-seq and cell experiment indicated that decreased FTL expression was related to cellular senescence, and this finding supported FTL as the biomarker of lung aging (Fig. 9g). Due to the expression pattern of FTL was discovered from the normal physiological process, using FTL as the recovery target of lung aging could have a high translational value and clinical safety.

## Discussion

Aging is the cumulative result of multiple factors, including the reduction of genomic stability, mitochondrial homeostasis, epigenetic modification changes, senescence-associated secretory phenotype, telomere shortening, telomerase activity decrease, and stem cell depletion[74,75]. In response to these mechanisms of aging, many studies have proposed various potential therapeutic strategies, such as telomerase reactivation[76], stem cell induction/transplantation[77], lifestyle intervention[78], and the development of universal anti-aging drugs[79]. However, the transition of anti-aging therapies from research to clinical practice has faced substantial challenges. For instance, some studies, such as the adeno-associated vector mediated telomerase reverse transcriptase protein expression[80], have not progressed beyond preclinical testing in animal models. Additionally, due to the long-term and heterogeneous problem of aging research, some clinical studies have encountered great difficulties, such as dietary restriction for anti-aging[81].

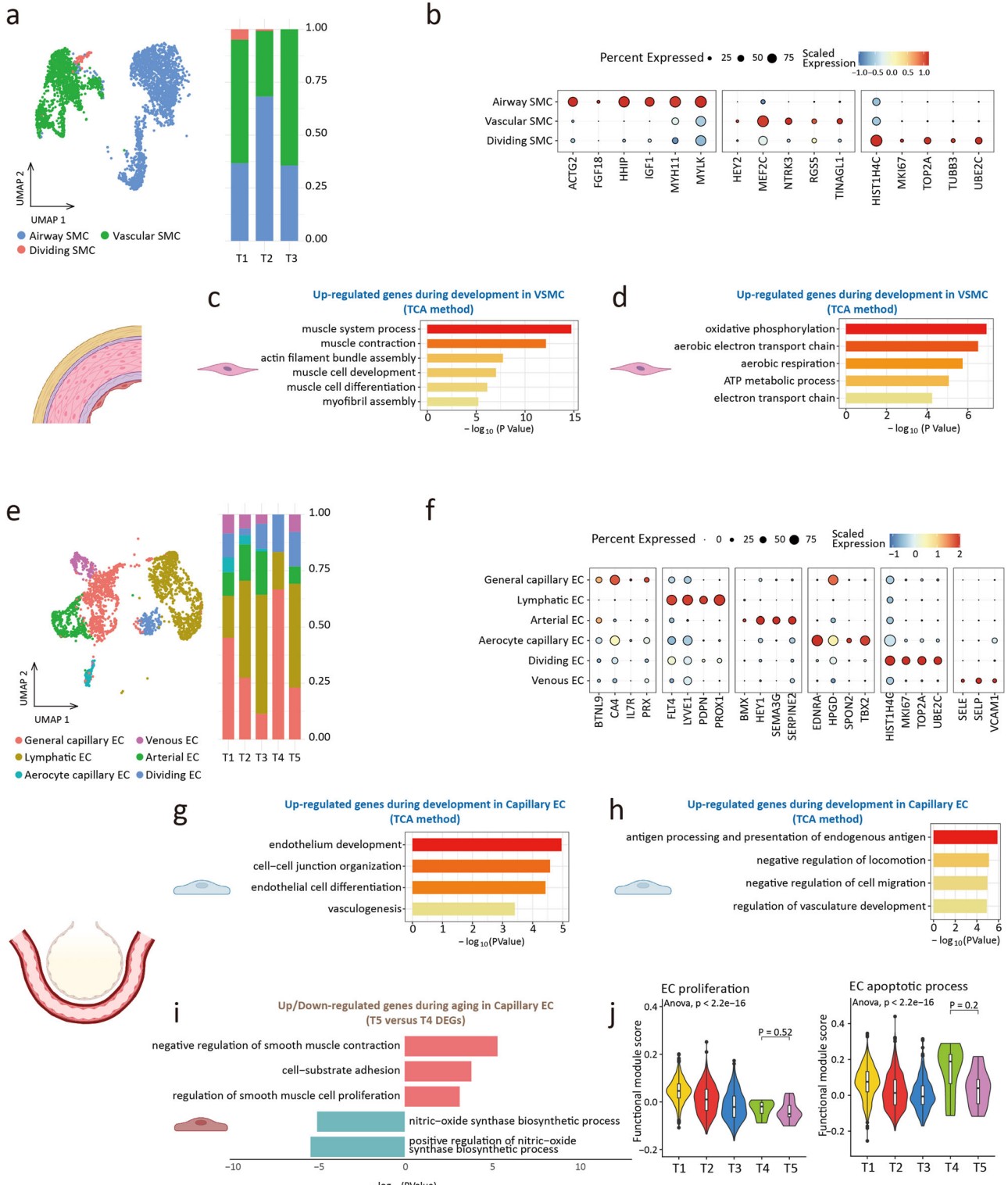

**Fig. 5 | Smooth muscle cell and endothelial cell. a** UMAP plots of smooth muscle cell (SMC) (left). Proportion of SMC subclusters in 3 groups (right). **b** Dot plots of specific marker genes of SMC. **c** Gene ontology enrichment analysis of gene expression in cluster 4 of VSMC (development). **d** Gene ontology enrichment analysis of gene expression in cluster 5 of VSMC (development). **e** UMAP plots of endothelial cell (EC) (left). Proportion of endothelial cell subclusters in 5 groups (right). **f** Dot plots of specific marker genes of endothelial cell. **g** Gene ontology enrichment analysis of gene expression in cluster 4 of capillary endothelial cell (development). **h** Gene ontology enrichment analysis of gene expression in cluster 6 of capillary endothelial cell (development). **i** Gene ontology enrichment analysis of up (top part, red)/down (bottom part, blue)-regulated gene expression in aging capillary endothelial cell. **j** The proliferative (GO: 0001935) scores and the apoptotic (GO: 0072577) scores of endothelial cells. The choice of ANOVA tests was based on the results of Shapiro–Wilk normality test. Dunnett tests were used to inter-group comparison.

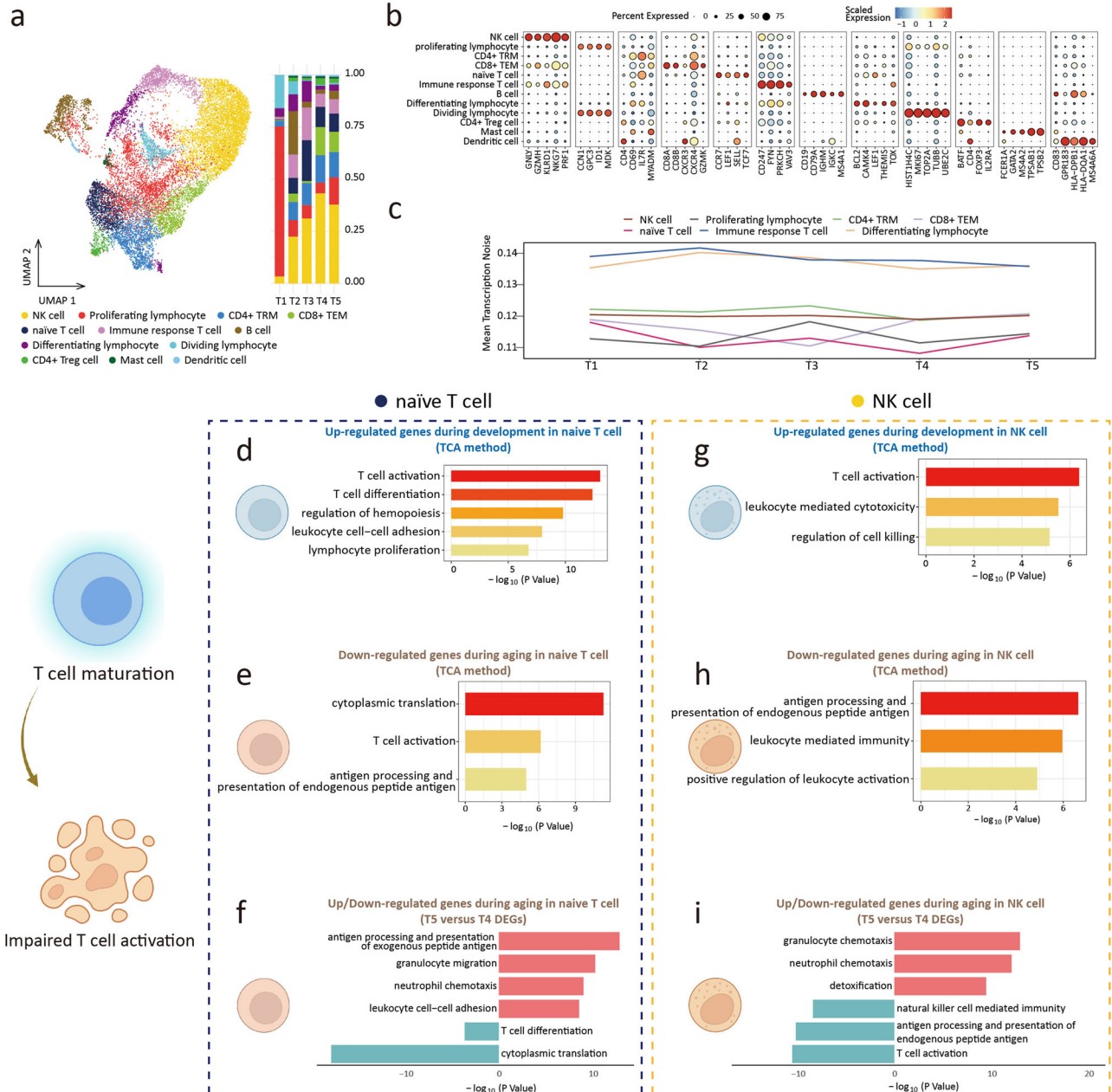

**Fig. 6 | Lymphoid cell. a** UMAP plots of Lymphoid cell (Lym) (left). Proportion of Lymphoid cell subclusters in 5 groups (right). **b** Dot plots of specific marker genes of Lym. **c** The age-associated changes of transcriptional noise of Lym subclusters. **d** Gene ontology enrichment analysis of gene expression in cluster 6 of naïve T cell (development). **e** Gene ontology enrichment analysis of gene expression in cluster 3 of naïve T cell (aging). **f** Gene ontology enrichment analysis of up (top part, red)/ down (bottom part, blue)-regulated gene expression in aging naïve T cell. **g** Gene ontology enrichment analysis of gene expression in cluster 2 of NK cell (development). **h** Gene ontology enrichment analysis of gene expression in cluster 6 of NK cell (aging). **i** Gene ontology enrichment analysis of up (top part, red)/down (bottom part, blue)-regulated gene expression in aging NK cell.

Moreover, interventions targeting the marker genes in aging may posed potential clinical safety concerns[82].

To explore more effective solutions for aging lungs, we used high-throughput omics data to observe the dynamic change of development and aging at the single-cell level of the lungs. We took advantage of the fact that human data enhanced the translational value, and the results of this study might be more consistent with the physiological rationale for identifying intervention targets from the normal life cycle, thus reducing the risk of clinical safety.

We identified dynamic transcriptional changes through a continuum atlas of the human lung during development to aging. For transcriptional noise, it has been theorized that transcriptional instability and increased transcriptional noise can cause cell fate drifts and lead to aging[83]. In 2019, Angelidis et al. found the increased transcriptional noise in the lung cells of aging mice[21]. Our study found that transcriptional noise was increased during aging, especially in AT2, FB, and neutrophils. We detected a common trend in epithelial cells during development, marked by a gradual decrease in transcriptional noise during development, a process related to the maturation of cell fate[84].

Additionally, our analysis revealed a dynamic functional enrichment in lung component cells, characterized by a gradual acquisition of specific cellular functions during development and loss during aging, such as surfactant homeostasis of AT2. This trend is consistent with previous findings in single-cell sequencing of the nervous system in mice[85].

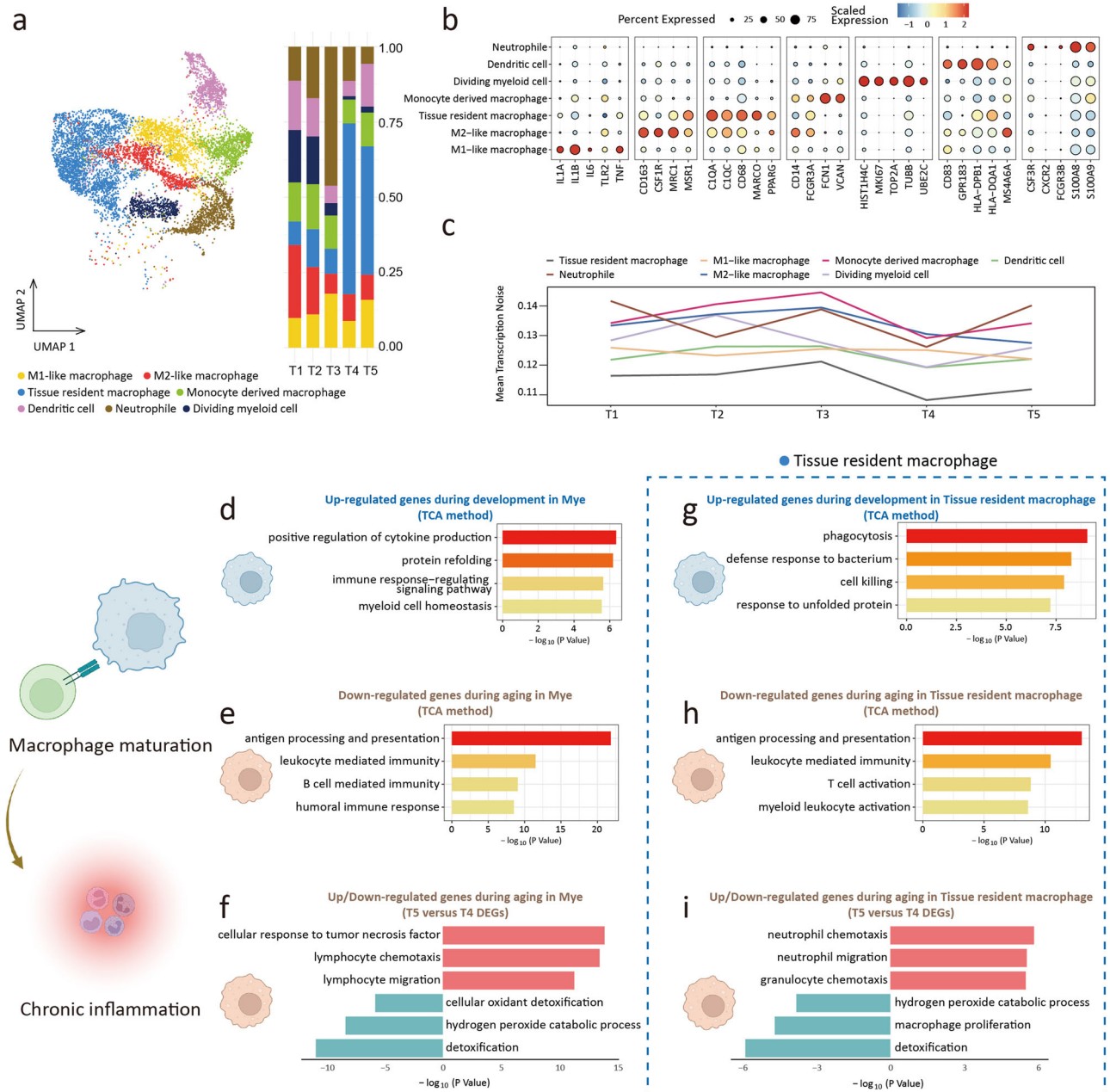

**Fig. 7 | Myeloid cell. a** UMAP plots of myeloid cell (Mye) (left). Proportion of Myeloid cell subclusters in 5 groups (right). **b** Dot plots of specific marker genes of Mye. **c** The age-associated changes of transcriptional noise of Mye subclusters. **d** Gene ontology enrichment analysis of gene expression in cluster 4 of Mye (development). **e** Gene ontology enrichment analysis of gene expression in cluster 6 of Mye (aging). **f** Gene ontology enrichment analysis of up (top part, red)/down (bottom part, blue)-regulated gene expression in aging Mye. **g** Gene ontology enrichment analysis of gene expression in cluster 1 of tissue-resident macrophage (development). **h** Gene ontology enrichment analysis of gene expression in cluster 4 of tissue-resident macrophage (aging). **i** Gene ontology enrichment analysis of up (top part, red)/down (bottom part, blue)-regulated gene expression in aging tissue-resident macrophage.

Moreover, we documented shifts in the relative proportions of cell subclusters throughout the human lifespan, characterized by stem cell population loss, such as Basal cells. Lung regeneration involves the activation of progenitor cells as well as cell replacement through the proliferation of remaining undamaged cells[86], which suggests that the embryonic lung has a relatively greater capacity for regeneration and repair[87].

Our results indicated a loss of immune homeostasis during the aging process, revealing a connection between immune cell dysfunction and chronic lung inflammation[88]. This overlap provides insights into the chronic inflammatory mechanisms in aging research[89].

To address the problems of lung aging, we found a key phenomenon in several cellular components of the lung. Expression of *FTL* increased in development and decreased in aging. As the component of ferritin, FTL regulates iron homeostasis in the cell[90], and dysregulated iron homeostasis is a hallmark of human aging[91]. It can be inferred from our results that the decrease of *FTL* expression in lung epithelial cells can impair cell homeostasis and induce cellular senescence. In addition to the function of *FTL* as an aging marker, restoring FTL expression levels has the potential to become an anti-aging treatment in the future. For example, Nodosin (a diterpenoid isolated from Isodon Serra) can be used to improve the expression of *FTL*[92], and this natural product has been shown to have anti-inflammatory and regulatory

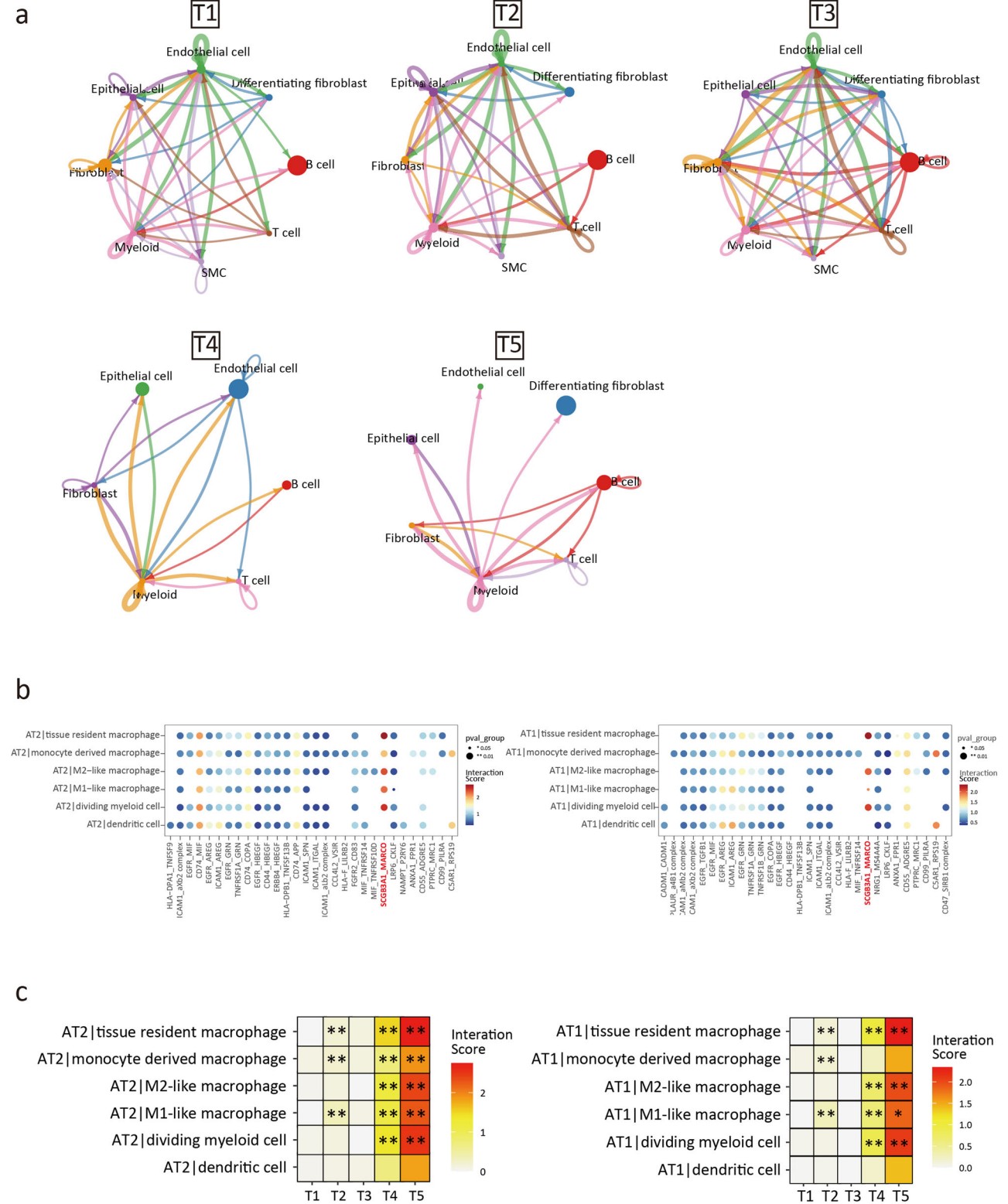

**Fig. 8 | Cell-cell interaction and pulmonary inflammation. a** Overall intercellular communication between main cell clusters (prob.cut.off = 0.3). **b** The interaction score between AT2 (left) / AT1 (right) and myeloid cell in T5 group. **c** The interaction score of *SCGB3A1* (ligand) -*MARCO* (receptor) pair between 5 groups. *$p < 0.05$, **$p < 0.01$.

effects on cell proliferation[93]. In addition to interfering with *FTL* expression, restoration of intracellular iron homeostasis using therapies, including iron chelation, also provides an option for intervention in lung aging[91,94].

Gender differences were included by default but not analyzed in this study. The sample size of last-trimester embryo individuals included in this study was relatively small.

## Methods
### Human lung specimens

The use of human lung tissue in this study was approved by the Human Ethics Committee of Fuwai Hospital, Chinese Academy of Medical Sciences. Adult lung specimens were collected with the informed consent of the patients, and aborted embryo specimens were obtained with the informed

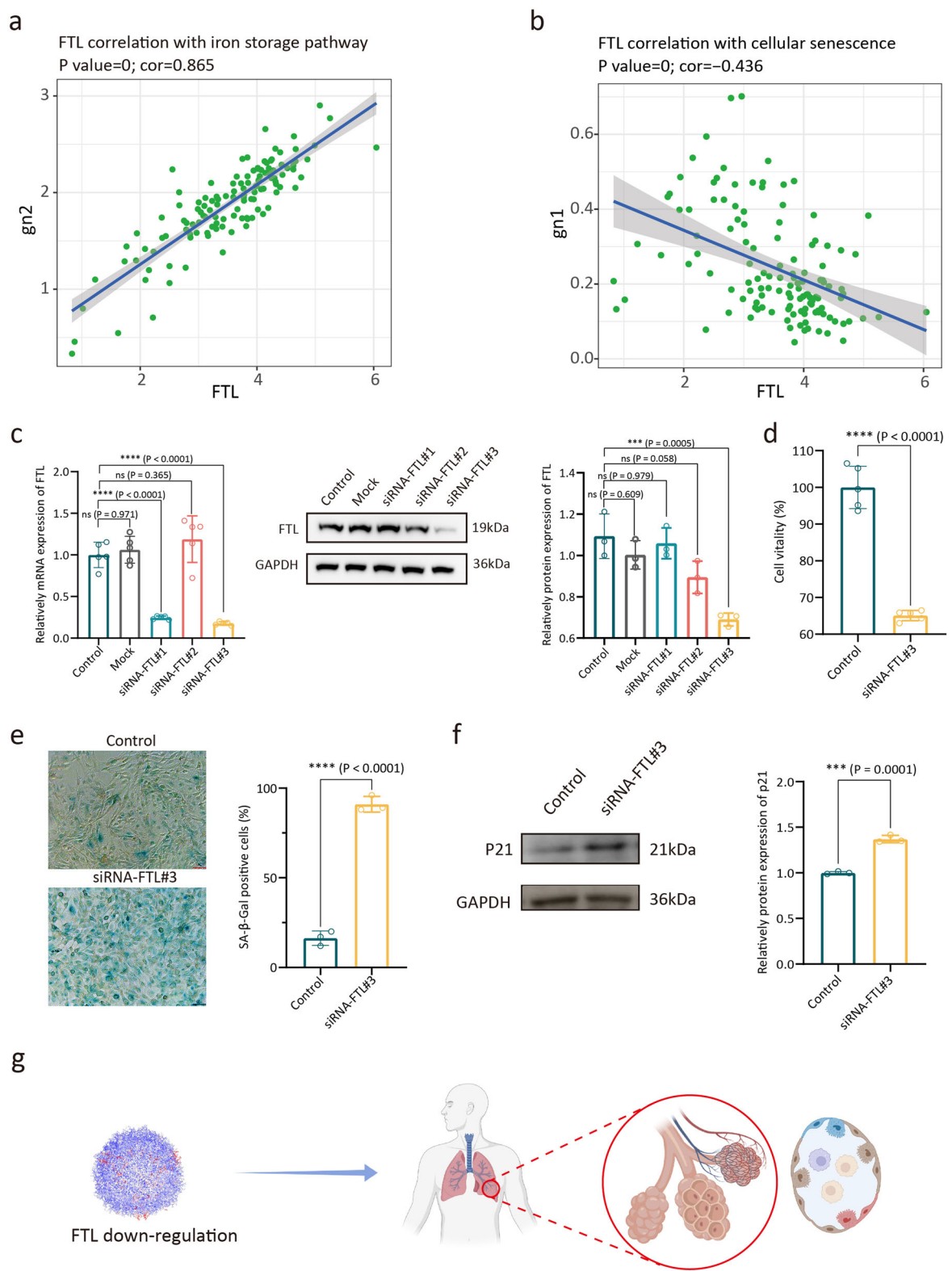

consent of the pregnant women. Embryonic lung tissue samples were collected from aborted embryos at 10, 12, 17, 20, 25, and 40 weeks of gestation. Adult lung tissue samples were collected from adults 47, 54, 67, 74 years of age who underwent lung surgery. These adult patients underwent surgery for lung nodules (<30 mm in diameter), and the samples were obtained from the normal tissue far away from the nodules that had been surgically removed.

Pulmonary nodules were reported as non-cancerous after surgery. All ethical regulations relevant to human research participants were followed.

**Collection of adult lung function parameters**
Static lung function monitoring data were retrospectively obtained from non-aging adults ($n = 18$) and aging adults ($n = 16$) in the third affiliated

**Fig. 9 | Down-regulation of FTL induced decreased iron homeostasis and cellular senescence. a** Correlations between the *FTL* expression level and the iron storage pathway (PW: 0000592) score of AT2 cell cluster. **b** Correlations between the *FTL* expression level and the cellular senescence (GO: 0090398) score of AT2 cell cluster. **c** RT-qPCR analysis of relatively mRNA expression levels of *FTL* in BEAS-2B cells after transfecting with siRNA, by ANOVA test (left), shown as the means ± SD. Western Blot (mid) and semi-quantitative analysis (right) of relatively protein expression of FTL in BEAS-2B cells after transfecting with siRNA, by ANOVA test, shown as the means ± SD. **d** The CCK8 assay showed the viability of BEAS-2B cells, by two-tailed *t*-test, shown as the means ± SD. **e** SA-β-gal activity was measured by X-gal staining, by two-tailed *t*-test, shown as the means ± SD. **f** Western Blot (left) and semi-quantitative analysis (right) of relatively protein expression of p21 in BEAS-2B cells after transfecting with siRNA, by two-tailed *t*-test, shown as the means ± SD. **g** Schematic drawing. The choice of ANOVA tests and two-tailed tests was based on the results of Shapiro-Wilk normality test. Dunnett tests were used to inter-group comparison. ***$p < 0.001$, ****$p < 0.0001$. For results in c and d, 5 replicates from different cell samples were used in each group. For results in (**e**, **f**), 3 replicates from different cell samples were used in each group.

hospital of Sun Yat-sen University. The age inclusion of the 2 groups matched the age of the T4 and T5 groups. Both groups of monitored individuals had no disease status or history of disease affecting lung function. These data were collected with the informed consent of the monitored individuals.

### Isolation of lung cells

Lung tissue samples were washed with phosphate buffer solution (PBS) and immersed in DMEM (Gibco, #11885084, USA) supplemented with 10% fetal bovine serum (Gibco, #10091148, USA). Samples were cut into small pieces. After washing with PBS, samples were digested in PBS containing 1000 U/mL Collagenase II (Worthington, #43J14367B, USA) at 37 °C for 15 min with gentle shaking. The above steps were repeated 3 times. The cell suspension was filtered with a 40 μm cell strainer (Falcon, #431750, USA) to get a single-cell suspension. Cells were collected by centrifugation at 400 *g*, 4 °C for 5 min. The supernatant was discarded. The cell pellets were re-suspended in DMEM containing 2% fetal bovine serum. The cell pellet was treated with 200 μL red blood cell lysis buffer (Beyotime, #C3702, China) for 10 min on ice. After centrifugation, the suspension was re-suspended. Single-cell suspension was harvested again with a 40 μm cell strainer.

### Single-cell library preparation and sequencing

Single-cell suspensions were loaded onto a Chromium Single Cell Controller (10x Genomics) to generate Gel Bead-In-Emulsions (GEMs). cDNA libraries were prepared using the single cell 5' solution v2 reagent kit (Chromium, 1000020) according to the protocol provided in the 10x Genomics Chromium Single Cell Immune Profiling Solution. After the reverse transcription step, droplets were disrupted and barcoded cDNAs were purified with DynaBeads, followed by 14 cycles of PCR amplification (98 °C for 45 s; [98 °C for 20 s, 67 °C for 30 s, and 72 °C for 1 min] × 14 cycles; 72 °C for 1 min). The resulting amplified cDNAs were sufficient to construct 5′ gene expression libraries. The cDNAs from single-cell transcriptomes (50 ng) were fragmented, subjected to 2 rounds of size selection with SPRI beads (avg. size 450 bp), and sequenced on the Illumina NextSeq platform (High Output V2 Kit, 150 cycles). All libraries were sequenced by an Illumina HiSeq 4000 sequencer.

### Sequencing data processing

Raw gene expression matrices were generated for each sample by the Cell Ranger (Version 6.1.2) Pipeline coupled with human reference version GRCh38-2020-A. The output-filtered gene expression matrices were analyzed by R software (Version 4.1.2) with the Seurat package (Version 4.1.1). A custom R script was used to combine the expression data and metadata from all libraries corresponding to a single batch, and cells with fewer than 200 Features were removed. The expression data matrix was filtered to retain genes with >5 UMI counts and then loaded into a Seurat object along with the library metadata for downstream processing. The percentage of mitochondrial transcripts for each cell (percent. mt) was calculated and added as metadata to the Seurat object. Cells were further filtered before dimensionality reduction (nGene-min. 200; percent. mt-max. 10%). Low-quality libraries identified were removed from the dataset. Expression values were then scaled to 10,000 transcripts per cell and Log-transformed. Effects of variable (percent. mito) were estimated and regressed out using a GLM (ScaleData function, model.use = ” linear”), and the scaled and centered residuals were used for dimensionality reduction and clustering. Before the

clustering, we first applied Canonical correlation analysis (CCA) implemented in Seurat to correct the batch effects among the experiments and integrate the gene expression matrix of all samples into a whole matrix.

### Dimensionality reduction

We used 2000 genes with high cell-to-cell variation, which were calculated using the FindVariableFeatures function in Seurat for further dimensionality reduction. To reduce the dimensionality of the datasets, the RunPCA function was conducted with default parameters on linear-transformation scaled data generated by the ScaleData function. Next, the ElbowPlot and DimHeatmap functions were used to identify the proper dimensions of each dataset.

### Cell clustering and identification of marker genes

After non-linear dimensional reduction and projection of all cells into two-dimensional space by UMAP, we initially built a graph of cells by using the K-Nearest Neighbours (KNN) algorithm applied to the PC-reduced space where each cell was connected to its 50 most similar cells using the manhattan distance. Then, to build the final graph of cells, the edge weight between any two cells was computed as the Jaccard similarity, i.e. the proportion of neighbors they share. The Louvain algorithm with a resolution parameter equal to 0.25 was used to find communities of cells in this graph. Differentially expressed genes in each cluster were identified by the findAllMarkers function of the Seurat package, which compares the expression of a gene in each cluster versus all the others by using the *t*-test.

### Reclustering of major cell types

To identify subtypes or cells in different states within a major cell type, we used a two-round clustering strategy. Firstly, cells belonging to a cell type were extracted from the normalized gene expression matrix of each sample and a combined gene expression matrix of all samples was prepared. Like we did on the whole dataset, variably expressed genes were identified by the FindVariableGenes function in Seurat. After PCA analysis, we selected the top PCs and performed clustering analysis using CCA.

### Differential expression genes identification and functional enrichment

Differential gene expression testing was performed using the FindMarkers function in Seurat with the parameter "test.use=t" by default, and the Benjamini-Hochberg method was used to estimate the false discovery rate (FDR). Enrichment analysis for the functions of the DEGs was conducted using clusterProfiler version 4.2.2 R package. "Biological Processes" gene ontology annotations for all molecules on the surface screen were compiled from org.Hs.eg.db(v3.14.0).

### Functional module analysis

We used cell scores to evaluate the degree to which individual cells expressed a certain predefined expression functional gene set. The cell scores were initially based on the average expression of the genes from the predefined gene set in the respective cell. For a given cell i and a gene set j(Gj), the cell score SCj(i) quantifies the relative expression of Gj in cell i as the average relative expression (Er) of the genes in Gj compared to the average relative expression of a control gene set (Gjcont): SCj(i) = average (Er (Gj, i)) −average (Er (Gjcount, i)). The control gene set was randomly selected based on aggregate expression levels bins, which yield a comparable

distribution of expression levels and oversize to that of the considered gene set. The AddModuleScore function in Seurat was used to implement the method with default settings. Detailed information on the gene sets can be found in Supplementary Table 2.

## Temporal patterns of time course data

We applied the R package TCseq (v1.18.0) to analyze the differentiation of experimental conditions. TCseq compares the temporal patterns of a gene between experimental conditions, taking into consideration all of the possible co-expression modules that this gene may participate in. By default, we use $k = 6$ to get the co-expression modules.

## Estimation of transcriptional noise

To ascertain the robustness of age-dependent transcriptional noise, for each time course measurement, we first divided the cells into cell types and computed the mean expression vector for each cell type. We then calculated the Euclidean distance between each cell and its corresponding cell type mean vector. The individual data points were summarized as boxplots. Finally, as an alternative method to obtain a measure of the transcriptional noise of a single cell, we selected a set of invariant genes evenly across the range of mean expression. First, we binned the genes in 10 equally sized bins by mean abundance, then we used these genes to determine the Euclidean distance from each cell to the average profile across all cells.

## Pseudotemporal ordering of cells

Monocle (v2.16.0) aims to resolve cellular transitions during differentiation through pseudotemporal profiling of scRNA-seq data. After inputting the cell-gene matrix into the "newCellDataSet" function with its clustering information, it was computed into a lower dimensional space based on the discriminative dimensionality reduction with trees (DDRTree) method, a more recent manifold learning algorithm, and then cells were ordered according to pseudotime.

## Transcriptional factor (TF) activity analysis

Transcription factor activity was analyzed using pySCENIC (v0.11.2) per cell type with raw count matrices as input. The regulons and TF activity (AUC) for each cell were calculated with the pySCENIC pipeline with motif collection version mc9nr. The differentially activated TFs of each subcluster were identified by the test against all the other cells of the same cell type.

## Overlap genes analysis

The differential expression of genes between cell types and sample groups was computed separately using the Seurat FindAllMarkers function. Genes with a filtered criterion of a $p$-value < 0.05 and an average log2 fold change > 0.25 were identified as representative genes for each group. Subsequently, within the databases encompassing hotspot genes from DisGeNET, we specifically identified a subset of Differentially Expressed Genes (DEGs) that exhibited overlap with genes associated with Pulmonary Hypertension (C0020542), Emphysema (C0034067), Chronic Obstructive Pulmonary Disease (COPD) (C0024117), and Asthma (C0004096).

## Cell-cell communication analysis

We applied the Cellphone database of known receptor-ligand pairs to assess cell-cell communication in our dataset. Genes from the Seurat object were renamed to Human gene names and then reformatted into the input format described on the CellphoneDB website. Cells were fed into the cellphonedb calculate program using 50 iterations, a precision of 3, and a 0.1 ratio of cells in a cluster expressing a gene. Then interactions were trimmed based on significant sites with $p < 0.05$.

## Correlation analysis

Signature scores of each AT2 were defined as the mean expression of gene signatures. Genes associated with 'iron storage pathway (PW: 0000592)', and 'cellular senescence (GO: 0090398)' were used to define the signature score. Cells in T4 and T5 groups with all scores upper than 0.01 were used to

calculate the Pearson correlation between normalized gene expression of "FTL"/"CISH" and each score.

## Immunofluorescence staining

About 5-μm-thick formaldehyde-fixed paraffin-embedded sections were prepared and followed by antigen retrieval with EDTA solution (pH 9.0, ZSBG-BIO, #ZLI-9068, China). After being blocked by goat serum (ZSBG-BIO, #ZLI-9021, China) for 1 h, the sections were incubated with primary antibodies overnight at 4°C. The next day, sections were incubated with fluorescence-labeled secondary antibodies for 1 h at room temperature, and then counterstained and mounted with DAPI (ZSGB-BIO, #ZLI-9557, China). Antibodies used for immunofluorescence staining were follows: anti-FTL (Proteintech, #10727-1-AP, China), anti-SFTPC (Abcam, #ab90716, GBR), anti-AGER (Abcam, #ab216329, GBR), anti-EEF1A1 (Proteintech, #11402-1-AP, China), and anti-VIM (Abcam, #ab8978, GBR). Fluorescence was observed under a ZEISS LSM800 confocal laser scanning microscope. The intensity of target gene expression was measured with Image-Pro Plus (Version 6.0, Media Cybernetics, USA).

## Masson's trichrome staining

Paraffin-embedded sections were deparaffinized and rehydrated. Sections were rinsed with distilled water and stained with potassium dichromate solution at RT overnight. Sections were then stained with iron hematoxylin working solution for 5 min, followed by staining with Ponceau-acid fuchsin solution for 5 min. Sections were incubated in the phosphomolybdic-phosphotungstic acid solution for 2 min, and stained in aniline blue solution for 2 min. After rinsed with distilled water, sections were dehydrated and mounted. Image-Pro Plus (Version 6.0, Media Cybernetics, USA) was used to semi-quantify the area ratio of fibrosis (blue).

## Cell culture and siRNA transfection

BEAS-2B cells were cultured with Epithelial Cell Medium (ScienCell, #3211, USA) according to the protocol of ATCC. BEAS-2B cells were seeded in the 12-well plates and were transfected 24 h later with siRNA-FTL using ribo-FECTTM CP Transfection Kit (RIBOBIO, #C10511-05, China) according to the manufacturer's protocol (The final concentration of siRNA-FTL−3 was 30 nmol/L).

## RT-qPCR analysis

Total tissue RNA was isolated by using TRIzol™ reagent (Invitrogen, #15596018, USA). cDNA was prepared from 500 ng of total RNA using a cDNA synthesis kit (TaKaRa, #RR036A, Japan). Real-time Quantitative PCR (RT-qPCR) was conducted with PowerUp™ SYBR™ Green Premix (ABI, #A25742, USA). Expression of FTL mRNA was calculated relative to that of 18 s rRNA. FTL: Forward primer 5'- CAGCCTGGTCAATTTGT ACCT-3', Reverse primer 5'-GCCAATTCGCGGAAGAAGTG-3'; 18 s rRNA: Forward primer 5'-CGGCTACCACATCCAAGGAA-3', Reverse primer 5'-GCTGGAATTACCGCGGCT-3'.

## Western Blot analysis

Total protein was extracted by using RIPA (Beyotime, #P0013, China), and then quantified using the BCA protein assay kit (ThermoFisher Scientific, #23227, USA). The protein solution was separated by SDS-PAGE (10%). After incubation with the blocking buffer, PVDF membranes were probed with primary antibodies, and subsequently with secondary antibody conjugated with horseradish peroxidase. The visualization of the Blot was achieved through chemiluminescence (Roche, #11500708001, Switzerland). Primary antibodies were as follows: anti-FTL (Proteintech, #10727-1-AP, China), anti-p21 (Proteintech, #10355-1-AP, China), and anti-GAPDH (GeneTex, #GTX100118, USA). The semi-quantitative analysis was realized by Image J software (National Institutes of Health, USA).

## Cell vitality assays

Cell Counting Kit-8 (CCK-8) (Beyotime, #C0038, China) was used to detect cell vitality. BEAS-2B cells were seeded in the 96-well plates. The cell vitality

rates were detected by multimode microplate reader (Tecan, #Infinite-M200, Swiss).

## Measurement of senescence-associated beta-galactosidase (SA-β-gal)

The activity of SA-β-gal of BEAS-2B cells was determined using Senescence β-Galactosidase Staining Kit (Beyotime, #C0602, China). SA-β-gal positive cells (blue color) were counted under microscope and expressed as percentage of total cells.

## Statistics and reproducibility

For non-scRNA-seq analysis, for 2-group comparisons, according to the characteristics of data normality and variance homogeneity, two-tailed *t*-test and Mann–Whitney test were used. The method of data normality test was the Shapiro-Wilk test. Multiple group comparisons were made by one-way ANOVA and Dunnett test. According to the characteristics of qualitative data distribution, Chi-square incorporating Yates' correction for continuity was used to compare qualitative variables across the groups of study subjects. Values with $P < 0.05$ were considered statistically significant. Normally distributed data were presented as Mean ± SD, and non-normally distributed data are presented as Median (IQR). This non-snRNA-seq statistical analysis was performed by SPSS software (Version 23.0, IBM Corp., USA) and GraphPad Prism (Version 8.3.1, GraphPad Software, USA). For snRNA-seq data, analysis was performed in R software, and statistical significance was accepted for $P < 0.05$.

## Reporting summary

Further information on research design is available in the Nature Portfolio Reporting Summary linked to this article.

## Data availability

The raw sequence data and processed expression matrix files have been deposited in the Gene Expression Omnibus (GSE: 260769). Any other data are available from the corresponding author on reasonable request.

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

## Acknowledgements
This work was supported by the National Natural Science Fund for Distinguished Young Scholars of China (82125004; to J.S.). Illustrations in Figs. 1a, 2a, 3a, 4a, d–f, 5c, g, i, 6a, 7a, d–i, 9g, and Supplementary Fig. 8e–g were created with BioRender.com (Invoice number 85D90E15-0005). Thanks to Yu Zhang (Ph.D. of Fuwai Hospital, Chinese Academy of Medical Sciences and Peking Union Medical College) for her suggestion on editing the main text for English language and grammar. Thanks to Yuanheng Huang (Ph.D. of the third affiliated hospital of Sun Yat-sen University, Sun Yat-sen University) for his contribution on clinical data collection.

## Author contributions
J. Song, X. Fu, and J. Zhang designed the study. H. Jia and Y. Chang processed and analyzed the scRNA-seq data. Y. Chen and Lei. H performed the cell experiment. H. Zhang, X. Hua, M. Xu, and Y. Sheng performed scRNA-seq experiment and sample collection. H. Jia, Y. Chang, X. Chen, N. Zhang, and H. Cui performed immunofluorecence staining experiment. H. Jia, Y. Chang, Y. Chen, and J. Song wrote and edited the manuscript. X. Chen and J. Song supervised scRNA-seq data collection and analysis. J. Song, X. Fu, and J. Zhang supervised the entire study.

## Competing interests
The authors declare no competing interests.
