## [Peer Review File · Communications Biology]

Reviewers' comments:

Reviewer #1 (Remarks to the Author):

The current study by Jia et al. used scRNA-Seq to profile lung cells during development and aging. They described various cellular and molecular changes in the lung during aging and highlighted alterations in epithelial, mesenchymal, endothelial, and immune cells. The report of a potential role of the FTL gene is interesting. Overall, the study is intriguing but is primarily descriptive, lacking confirmatory experiments, and failing to provide a solid conclusion to the scientific community.

1-Given that the study focuses on aging, the paper is missing data from the middle age group (20-30 years old), and they could enhance their analysis by incorporating publicly available data along with their own to present a more comprehensive narrative.

2-The discussion is very general and lacks depth, as well as a concise conclusion. Additionally, the author failed to discuss other reported studies focused on aging and compare their findings with them.

3-The description of different cell types in the lung using almost the same analysis tools without a deep analysis and discussion of specific cell types makes the paper lack depth and conciseness.

4-The sex of the individuals in the study is not included. The limitation must be acknowledged, and a discussion of results must be provided in this context.

5-The T3 group, with an n=1 sample size, is statistically small, which weakens the ability to draw robust conclusions.

6-The gene expression in the violin plots is not clear regarding whether it is statistically significant, as all the violin plots in the figures lack the statistical analysis. It should be included to clarify whether the differences are statistically significant or not.

7-The authors highlight the finding regarding FTL gene, but there is no introductory information provided about the gene's expression or whether it has a known role in the lung or other organs.

8-Subclustering of total epithelial cells can be misleading. It would be better to subcluster the airway and alveolar epithelial cells separately, as this approach would provide a more in-depth analysis for the study.

9-Additionally, the author only highlights the changes in ciliated cells, but they did not discuss the changes in basal or club cells, which are two important cell types present in lung airways. Especially noteworthy are basal cells, as they serve as the stem cells of lung airways and play a crucial role in the repair and regeneration of airway epithelium.

10-The expression of the EEF1A1 gene in AT1 cells is not clear using immunofluorescence analysis, as the authors did not use an AT1-specific marker to confirm this finding.

11-The transcription shift between different groups needs to be included in the figures and discussed in the text.

12-It is better to use different colors than green and yellow in immunofluorescence images, as green and yellow are not appropriate and can be confusing, making it very difficult to distinguish the co-expression of the two proteins.

13-Overall, immunofluorescence images are not clear enough to convey the message that the author wants to highlight and need to be quantified.

14-In Fig. 3h, it is clear that there is a difference in collagen deposition between T4 and T5, but the authors failed to discuss this difference in detail in the text. Please provide a more detailed discussion.

15-As the study is about aging, and the authors reported an increase in collagen deposition in the aged samples, they did not include a discussion or link to lung fibrosis, which is considered an age-associated disease.

16-It has been reported that one of the most complicated aspects of aging is losing endothelial cell identity and the associated vascular network, leading to hypoxic conditions in the aged lung. The authors need to discuss this in relation to their findings.

17-The expression of the FTL gene is not clear; it is uncertain whether it exhibits cell-specific expression or if it is ubiquitously expressed in all lung cells.

18-The validation and quantification of FTL gene expression are not clear. It would be better if the author used qPCR analysis alongside the IF.

19-In the figures, the schematic diagrams lack detailed information, making it difficult to follow.

20-Many axes in the figures do not have titles.

21-The paper needs to be spell-checked, as there are errors.

Reviewer #2 (Remarks to the Author):

The authors address a very timely issue and create a valuable dataset. This manuscript is of interest to lung researchers and researchers interested in aging in general and the lung specifically.

However, the data is underused and a lot of the conclusions drawn solely rely on computational analysis and prediction without any experimental validation. The authors link their findings to functional impairments in the aged lung. This is however pure correlation to previous findings. A lot of the

conclusions need to be toned down .

One of the reasons for this is that the amount of single lungs is very low. This is understandable but at the same time presents a major drawback for conclusions since they are based on 2 lungs for one age group. Integration of publically available data from the human lung cell atlas could help with that

1. It would be of value to understand global changes in cell type proportions during aging and identify and describe different patterns (increase/decrease/increase and the decrease) and identify common regulatory pathways?
2. Are any of the cell types/ neighborhoods significantly changed with aging?
3. Can the authors correlate any of their findings to functional parameters in their cohort?
4. The confirmation on IF of single cell findings is appreciated, however, lower magnifications and quantifications are needed.
5. The authors should comment on recently identified pathogenic cell states in lung pathologies including in fibrosis and investigate their occurrence in normal lung aging
6. The authors have to expand their analysis of cellular senescence beyond the measurement of SA- β -Gal
7. The quality of the figures is really low, eg Figure 1

Reviewer #3 (Remarks to the Author):

Hoa Jia and colleagues conducted in-depth single-cell sequencing on human lungs at different stages of development (fetal lungs) and throughout the ageing process. Their article is highly relevant to the field, as it described changes in the molecular and cellular landscape over time. However, despite the amount of data presented, the manuscript is mainly descriptive and relies on the interpretation of the single-cell sequencing results.

Major Comments:

1. What are the lung function and health status of the donors whose tissue was used for the non-ageing and ageing adult groups? Have these individuals experienced a decline in lung function due to ageing?
2. Could the gene expression patterns identified be related to the onset of age-related diseases such as emphysema and COPD? For instance, studying a 74-year-old individual with a healthy lung could provide insight into the cellular and molecular mechanisms responsible for maintaining lung structure and function during the ageing process.
3. It is recommended that the authors expand the discussion chapter to include more of the results that were described but not discussed (e.g. changes in immune function and cell proportion of lymphocytes; changes in the immune and antigen presentation function of myeloid cells and etc.). Doing so will provide a better understanding of how these observations relate to lung ageing and age-related diseases.
4. The authors should consider excluding the following sentence from the abstract: We believed that intervening in the expression of FTL can resist pulmonary ageing through iron homeostasis mechanism. The results show that silencing of the ferritin light chain (FTL) may induce cellular senescence in vitro.

However, this is far from the proposed intervention targeting FTL as anti-ageing therapy. Moreover, the authors should consider distinguishing the concept of ageing and cellular senescence.

5.

Minor Comments:

1. The authors should consider using well-accepted abbreviations when referring to different cell types instead of “Fibs, Epis, Neu, Ery and Endos”.
2. The figures and panel labelling may be difficult to read when adapted to A4 page.
3. The authors may consider spelling out the abbreviations that appear in a figure in the corresponding figure legend.

Dear Reviewers,

We have revised our manuscript according to the comments. The parts marked in yellow are content modifications, and the green marks are grammatical or sentence readability adjustments. In addition, considering the contribution of Prof. Xiaodong Fu to the analysis and the content when we revised the manuscript, we would like to list him as the corresponding author.

The contents of the two supplements, “Statistics and Reproducibility” and “Data availability statement”, were revised according to the submission request and were also marked in yellow.

In terms of data availability, we will upload the raw data before you plan to accept this manuscript or need the data for review, and we will upload it to a public database.

Thank you for your help with our manuscript.

Comments of Reviewer1

Comment 1: Given that the study focuses on aging, the paper is missing data from the middle age group (20-30 years old), and they could enhance their analysis by incorporating publicly available data along with their own to present a more comprehensive narrative.

Answer 1: Thank you for your comment. Based on your suggestion, we found the data sets from middle-aged lung samples and tried to incorporate the data sets into our analysis (Nature. 2022 Apr;604(7904):111-119) (GSM5388412 Donor 2_RNA-seq and GSM5388413 Donor 3_RNA-seq). However, our scRNA-seq libraries were prepared by using the single cell 5' solution v2 reagent kit, and these data sets from middle-aged lung samples were prepared by using the 3' solution single index kit. The heterogeneity of the data produced by these two different sequencing protocols limits the combined analysis of the data sets (Genome Med. 2017 Aug 18;9(1):75). We tried to do a follow-up analysis where we repeated almost all of the analysis processes involved in the Version 1 manuscript, but the results suggested that the two scRNAseq data sets were too heterogeneous from our data due to the reagents used at the time the libraries were established.

Respond Figure 1. PCA analysis and Pearson correlation analysis of samples.

a. PCA analysis; **b.** Pearson correlation heatmap. The heterogeneity of external data sets can be found through these two analysis methods.

Change 1: -.

Comment 2: The discussion is very general and lacks depth, as well as a concise conclusion. Additionally, the author failed to discuss other reported studies focused on aging and compare their findings with them.

Answer 2: Thank you for your comment. We have optimized and discussed in depth of both our "Results" and "Discussion" sections according to your suggestions.

Change 2: "Results" and "Discussion". Marked in YELLOW.

Comment 3: The description of different cell types in the lung using almost the same analysis tools without a deep analysis and discussion of specific cell types makes the paper lack depth and conciseness.

Answer 3: Thank you for your comment. According to your suggestions, we have selected different analysis methods for different cell types. Among them, we mainly focus on the analysis of epithelial cells, and the subsequent experimental verification part is also aimed at the findings in epithelial cells.

Change 3: Section "Dynamic changes of pulmonary epithelial cells during development and aging" and Figure 3, marked in YELLOW.

Comment 4: The sex of the individuals in the study is not included. The limitation must be acknowledged, and a discussion of results must be provided in this context.

Answer 4: Thank you for your comment. We added this statement in "limitations".

Change 4: Section "Limitations". Marked in YELLOW.

Comment 5: The T3 group, with an n=1 sample size, is statistically small, which weakens the ability to draw robust conclusions.

Answer 5: Thank you for your comment. We added this statement in "limitations".

Change 5: Section "Limitations". Marked in YELLOW.

Comment 6: The gene expression in the violin plots is not clear regarding whether it is statistically significant, as all the violin plots in the figures lack the statistical analysis. It should be included to clarify whether the differences are statistically significant or not.

Answer 6: Thank you for your comment. We have revised all the violin plots according to your suggestion.

Change 6: Figure 2, 4, 5, and figure s6, s8.

Comment 7: The authors highlight the finding regarding FTL gene, but there is no introductory information provided about the gene's expression or whether it has a known role in the lung or other organs.

Answer 7: Thank you for your comment. We have added the description of FTL according to your suggestion.

Change 7: The 4th paragraph of "Dynamic changes of pulmonary epithelial cells during development and aging", the 1st paragraph of "Decreased FTL expression induced cellular senescence", and the 4th paragraph of "Discussion". Marked in YELLOW.

Comment 8: Subclustering of total epithelial cells can be misleading. It would be better to subcluster the airway and alveolar epithelial cells separately, as this approach would provide a more in-depth analysis for the study.

Answer 8: Thank you for your comment. We have re-clustered and analyzed epithelial cells according to your suggestion.

Change 8: Section “Dynamic changes of pulmonary epithelial cells during development and aging” and Figure 2, 3, marked in YELLOW.

Comment 9: Additionally, the author only highlights the changes in ciliated cells, but they did not discuss the changes in basal or club cells, which are two important cell types present in lung airways. Especially noteworthy are basal cells, as they serve as the stem cells of lung airways and play a crucial role in the repair and regeneration of airway epithelium.

Answer 9: Thank you for your comment. We have added the discussion on the basal cells and club cells.

Change 9: Section “Dynamic changes of pulmonary epithelial cells during development and aging” and Figure 3, marked in YELLOW.

Comment 10: The expression of the EEF1A1 gene in AT1 cells is not clear using immunofluorescence analysis, as the authors did not use an AT1-specific marker to confirm this finding.

Answer 10: Thank you for your comment. We have revised this part according to your suggestion.

Change 10: Figure 2o and figure s4c.

Comment 11: The transcription shift between different groups needs to be included in the figures and discussed in the text.

Answer 11: Thank you for your suggestion. We have revised this part according to your suggestion.

Change 11: Figure 2c, 4c, 6c, 7c, and related results.

Comment 12: It is better to use different colors than green and yellow in immunofluorescence images, as green and yellow are not appropriate and can be confusing, making it very difficult to distinguish the co-expression of the two proteins.

Answer 12: Thank you for your comment. We have revised this part by using Red/White and Green in immunofluorescence images.

Change 12: Figure 2 (k, m, o), Figure 4k, figure s4 (a, b, c), and figure s6c.

Comment 13: Overall, immunofluorescence images are not clear enough to convey the message that the author wants to highlight and need to be quantified.

Answer 13: Thank you for your comment. We repeated the experiment to optimize the immunofluorescence image and quantified it.

Change 13: Figure s4 and figure s6c.

Comment 14: In Fig. 3h, it is clear that there is a difference in collagen deposition between T4 and T5, but the authors failed to discuss this difference in detail in the text. Please provide a more detailed discussion.

Answer 14: Thank you for your comment. We have discussed the difference in detail according to your suggestion.

Change 14: The 2nd paragraph of “Phenotypic changes of fibroblasts during development and aging”, marked in YELLOW.

Comment 15: As the study is about aging, and the authors reported an increase in collagen deposition in the aged samples, they did not include a discussion or link to lung fibrosis, which is considered an age-associated disease.

Answer 15: Thank you for your comment. We have added the discussion of FB phenotypic changes according to your suggestion.

Change 15: The 2nd paragraph of “Phenotypic changes of fibroblasts during development and aging”, marked in YELLOW.

Comment 16: It has been reported that one of the most complicated aspects of aging is losing endothelial cell identity and the associated vascular network, leading to hypoxic conditions in the aged lung. The authors need to discuss this in relation to their findings.

Answer 16: Thank you for your comment. We have added the description of Endothelial cell identity according to your suggestion.

Change 16: The 3rd and 4th paragraphs of “Dynamic changes of smooth muscle cells and endothelial

cells”, marked in YELLOW.

Comment 17: The expression of the FTL gene is not clear; it is uncertain whether it exhibits cell-specific expression or if it is ubiquitously expressed in all lung cells.

Answer 17: Thank you for your comment. As we describe in the manuscript, the regular change tendency of FTL expression is common in Epi and FB. When we included all lung cells, we also found a similar trend at the pseudo-bulk level.

Respond Figure 2. Relatively expression of FTL in pseudo-bulk level in single-cell analysis.

Change 17: -

Comment 18: The validation and quantification of FTL gene expression are not clear. It would be better if the author used qPCR analysis alongside the IF.

Answer 18: Thank you for your comment. We have added the experiment according to your suggestion.

Change 18: figure 3f.

Comment 19: In the figures, the schematic diagrams lack detailed information, making it difficult to follow.

Answer 19: Thank you for your suggestion. We have added the detailed information of schematic diagrams according to your suggestion.

Change 19: Figure 2-7

Comment 20: Many axes in the figures do not have titles.

Answer 20: Thank you for your comment. We have added the axes in our figures according to your suggestion.

Change 20: Figure 1-9 and figure s1-s11

Comment 21: The paper needs to be spell-checked, as there are errors.

Answer 21: Thank you for your comment. We have re-checked our manuscript and revised these errors.

Change 21: Marked in GREEN.

Comments of Reviewer2

Comment 1: It would be of value to understand global changes in cell type proportions during aging and identify and describe different patterns (increase/decrease/increase and the decrease) and identify common regulatory pathways?

Answer 1: Thank you for your comment. We have added the description in all the sections of Results according to your suggestion.

Change 1: Figure 2a, 4a, 5a, 5e, 6a, 7a, and their related results, marked in YELLOW.

Comment 2: Are any of the cell types/ neighborhoods significantly changed with aging?

Answer 2: Thank you for your comment. We have added the discussion of the changes of cell types during aging.

Change 2: Figure 2a, 4a, 5a, 5e, 6a, 7a, and their related results, marked in YELLOW.

Comment 3: Can the authors correlate any of their findings to functional parameters in their cohort?

Answer 3: Thank you for your comment. We have added this data according to your suggestion.

Change 3: Table 1 and the 2nd paragraph of "Phenotypic changes of fibroblasts during development and aging", marked in YELLOW.

Comment 4: The confirmation on IF of single cell findings is appreciated, however, lower magnifications and quantifications are needed.

Answer 4: Thank you for your comment. We have added the lower magnifications and quantifications according to your suggestion.

Change 4: Figure s4a, b, c, figure s6c, Figure 2k, m, o and Figure 4k.

Comment 5: The authors should comment on recently identified pathogenic cell states in lung pathologies including in fibrosis and investigate their occurrence in normal lung aging

Answer 5: Thank you for your comment. We have added the description according to your suggestion.

Change 5: The 2nd paragraph of “Phenotypic changes of fibroblasts during development and aging”, marked in YELLOW.

Comment 6: The authors have to expand their analysis of cellular senescence beyond the measurement of SA- β -Gal

Answer 6: Thank you for your comment. We have added the expression of p21 (a senescence marker) to support our findings.

Change 6: Figure 9f.

Comment 7: The quality of the figures is really low, eg Figure 1

Answer 7: Thank you for your comment. We have adjusted the layout and font size of all figures to ensure that readers will not be confused with reading and recognition. In addition, if needed, we can upload images in Adobe Illustrator format and PDF format, which can provide higher definition compared to JPG format.

Change 7: -

Comments of Reviewer3

Comment 1: What are the lung function and health status of the donors whose tissue was used for the non-ageing and ageing adult groups? Have these individuals experienced a decline in lung function due to ageing?

Answer 1: Thank you for your comment. These patients had undergone surgery for small lung nodules (less than 30mm in diameter), and we collected the removed tissue away from the lesion.

These nodules later proved to be non-cancerous. The donor had no history of chronic lung disease. We have added this information in the METHODS.

Change 1: Section "Human lung specimens", marked in YELLOW.

Comment 2: Could the gene expression patterns identified be related to the onset of age-related diseases such as emphysema and COPD? For instance, studying a 74-year-old individual with a healthy lung could provide insight into the cellular and molecular mechanisms responsible for maintaining lung structure and function during the ageing process.

Answer 2: Thank you for your comment. We have added the cohort data and related single-cell analysis according to your comment.

Change 2: Table 1 and figure s10d.

Comment 3: It is recommended that the authors expand the discussion chapter to include more of the results that were described but not discussed (e.g. changes in immune function and cell proportion of lymphocytes; changes in the immune and antigen presentation function of myeloid cells and etc.). Doing so will provide a better understanding of how these observations relate to lung ageing and age-related diseases.

Answer 3: Thank you for your comment. We have optimized and discussed in depth of both our "Results" and "Discussion" sections according to your suggestions.

Change 3: "Results" and "Discussion". Marked in YELLOW.

Comment 4: The authors should consider excluding the following sentence from the abstract: We believed that intervening in the expression of FTL can resist pulmonary ageing through iron homeostasis mechanism. The results show that silencing of the ferritin light chain (FTL) may induce cellular senescence in vitro. However, this is far from the proposed intervention targeting FTL as anti-ageing therapy. Moreover, the authors should consider distinguishing the concept of ageing and cellular senescence.

Answer 4: Thank you for your comment. We have revised this part according to your suggestion.

Change 4: In the **Abstract**.

Comment 5: The authors should consider using well-accepted abbreviations when referring to different cell types instead of “Fibs, Epis, Neu, Ery and Endos”.

Answer 5: Thank you for your comment. We have revised the abbreviations in our manuscript according to your suggestion. Fibroblast - FB; Epithelial cell - Epi; Neural cell - NC; Endothelial cell - EC;

Change 5: -

Comment 6: The figures and panel labelling may be difficult to read when adapted to A4 page.

Answer 6: Thank you for your comment. We have adjusted the layout and font size of all figures to ensure that readers will not be confused with reading and recognition. In addition, if needed, we can upload images in Adobe Illustrator format and PDF format, which can provide higher definition compared to JPG format.

Change 6: Figure 1-9 and figure s1-s11.

Comment 7: The authors may consider spelling out the abbreviations that appear in a figure in the corresponding figure legend.

Answer 7: Thank you for your comment. We have spelled out the abbreviations in our figure legends.

Change 7: The figure legends of Figure 1-9 and figure s1-s11.

REVIEWERS' COMMENTS:

Reviewer #1 (Remarks to the Author):

The authors have been responsive to previous reviews and have addressed many points. However, I recommend that the color scheme and overall appearance of the immunofluorescence images be modified to enhance co-localization and improve visual representation. Additionally, I suggest including the number of experiments and/or analyzed single cells in the figure legends for each panel. Furthermore, the resolution of all figures needs to be significantly enhanced.

Reviewer #2 (Remarks to the Author):

The authors addressed many of my concern, however some of the newly added data seems overinterpreted:

What do the authors mean with decreased transcriptional imaging (417)

What do the authors mean with "which suggests that the cellular fate characteristics of the embryonic lung can be used for lung repair after aging or disease" (428)

The authors cannot conclude that "These phenotypic changes led to a decrease in lung compliance" (214).

The authors have not addressed if there is any statistical significant difference in cell types/neighborhoods

There are a couple of typos that need to be fixed, eg. BEAS 2B cells
Gender differences are not included in the study- they were included by default but not analyzed- consider rephrasing

Reviewer #3 (Remarks to the Author):

The authors addressed all the concerns.

Dear reviewers,

Thank you for your help with our manuscript. We made the changes as suggested. We did not remove the yellow and green markers in revised version1. The content changes in this version 2 are marked in red.

In addition, we uploaded all the raw data, which are available in the Gene Expression Omnibus (GEO) database under the number GSE260769 (Release data: Mar 10, 2024).

If you are not satisfied with the resolution of some figures, please contact us, we can upload PDF format.

Series GSE260769	
Status	Private until Mar 10, 2024
Title	Dynamic changes of lung homeostasis from development-to-aging single-cell atlas
Organism	Homo sapiens
Experiment type	Expression profiling by high throughput sequencing
Summary	Aging is a global problem, in which lung aging is accompanied by functional decline and structural disorders, disturbing the health of the elderly population. To explore anti-aging methods, we constructed a dynamic atlas of 45 cell types, from embryonic development to aging on human lung samples, and we hoped to use the discoveries of development to solve the problems of aging. During development and aging, epithelial and immune cells underwent a process of acquisition and loss of obligatory function. During aging, we identified cellular phenotypic changes that result in decreased pulmonary compliance and immune homeostasis. Furthermore, we found a characteristic expression pattern for the ferritin light chain (FTL) gene, which was regulated upward during development and downward during aging in multiple component cells of the lung.
Overall design	The use of human lung tissue in this study was approved by the Human Ethics Committee of Fuwai Hospital, Chinese Academy of Medical Sciences. Adult lung specimens were collected with the informed consent of the patients, and aborted embryo specimens were obtained with the informed consent of the pregnant women. Embryonic lung tissue samples were collected from aborted embryos at 10, 12, 17, 20, 25, and 40 weeks of gestation. Adult lung tissue samples were collected from adults 47, 54, 67, and 74 years of age who underwent lung surgery. These adult patients underwent surgery for lung nodules (less than 30mm in diameter), and the samples were obtained from the normal tissue far away from the nodules that had been surgically removed. Pulmonary nodules were reported as non-cancerous after surgery.
Contributor(s)	Jia H , Chang Y , Chen Y , Chen X , Zhang H , Hua X , Xu M , Sheng Y , Zhang N , Cui H , Han L , Zhang J , Fu X , Song J

Reviewer 1:

Comment: The authors have been responsive to previous reviews and have addressed many points. However, I recommend that the color scheme and overall appearance of the immunofluorescence images be modified to enhance co-localization and improve visual representation. Additionally, I suggest including the number of experiments and/or analyzed

single cells in the figure legends for each panel. Furthermore, the resolution of all figures needs to be significantly enhanced.

Answer: Thank you for your comment.

- 1) For the color scheme of immunofluorescence images, we have optimized it in the modified version. For some parts that may have color matching problems (AGER-EEF1A1-DAPI), we have tried all combinations in the color panel, which should be optimal at present. In addition, we performed immunohistochemical pretesting before immunofluorescence to ensure that the targeting of the antibodies, particularly cell-labeled antibodies, was as expected.

- 2) For the number of experiments and/or analyzed single cells in the figure legends. We have added the number of experiments in the Figure 9 legend. For the number of single cells, we added a supplementary material with cell numbers, for both major cell types and subclusters.
- 3) For the resolution of all figures, we can upload all images in PDF format, which we believe can be used to provide higher resolution.

Change: Figure 9 legend. Marked in Red. Supplementary material_cellNumber

Reviewer 2:

Comment 1: What do the authors mean with decreased transcriptional imaging (417)

Answer 1: Thank you for your comment. It was a clerical error and we corrected it

Change 1: The 3rd paragraph of "Discussion". Marked in Red.

Comment 2: What do the authors mean with "which suggests that the cellular fate

characteristics of the embryonic lung can be used for lung repair after aging or disease" (428)

Answer 2: Thank you for your comment. As you suggested, we agreed that this part of the description was overinterpreted and we have revised it.

Change 2: The 3rd paragraph of "Discussion". Marked in Red.

Comment 3: The authors cannot conclude that "These phenotypic changes led to a decrease in lung compliance" (214).

Answer 3: Thank you for your comment. We have revised our description in response to your suggestion.

Change 3: The 2nd paragraph of "Phenotypic changes of fibroblasts during development and aging". Marked in RED.

Comment 4: The authors have not addressed if there is any statistical significant difference in cell types/ neighborhoods.

Answer 4: Thank you for your comment. Your suggestion was considered in the revision of our first edition, but given some of our sample-size limitations (as stated in "Limitations"), we thought that the description of statistical significance may be misleading to readers, and we chose to describe trends rather than statistical significance. To address this issue, we added the supplementary material for cell numbers.

Change 4: Supplementary material_cellNumber.

Comment 5: There are a couple of typos that need to be fixed, eg. BEAS 2B cells.

Answer 5: Thank you for your comment. We have corrected these typos according to your suggestion.

Change 5: Marked in RED.

Comment 6: Gender differences are not included in the study- they were included by default but not analyzed-consider rephrasing

Answer 6: Thank you for your comment. We have revised our description according to your suggestion.

Change 6: The last paragraph of “Discussion”. Marked in Red.

Reviewer 3:

Comment 1: The authors addressed all the concerns.

Answer 1: Thank you for your help with our manuscript. Your suggestions are of great significance for the improvement of the quality of our manuscript. We are honored to meet your evaluation criteria.

Change 1: -